# A classification algorithm for selective dynamical downscaling of precipitation extremes

Edmund P. Meredith[1], Henning W. Rust[1], and Uwe Ulbrich[1]

[1]Institut für Meteorologie, Freie Universität Berlin, Carl-Heinrich-Becker-Weg 6-10, D-12165 Berlin, Germany

*Correspondence to:* Edmund P. Meredith (edmund.meredith@met.fu-berlin.de)

**Abstract.** High-resolution climate data $O(1\ km)$ at the catchment scale can be of great value to both hydrological modellers and end users, in particular for the study of extreme precipitation. While dynamical downscaling with convection-permitting models is a valuable approach for producing quality high-resolution $O(1\ km)$ data, its added value can often not be realised due to the prohibitive computational expense. Here we present a novel and flexible classification algorithm for discriminating between days with an elevated potential for extreme precipitation over a catchment and days without, so that dynamical downscaling to convection-permitting resolution can be selectively performed on high-risk days only, drastically reducing total computational expense compared to continuous simulations; the classification method can be applied to climate model data or reanalyses. Using observed precipitation and the corresponding synoptic-scale circulation patterns from reanalysis, characteristic extremal circulation patterns are identified for the catchment via a clustering algorithm. These extremal patterns serve as references against which days can be classified as potentially extreme, subject to additional tests of relevant meteorological predictors in the vicinity of the catchment. Applying the classification algorithm to reanalysis, the set of potential extreme days (PEDs) contains well below 10 % of all days, though includes essentially all extreme days; applying the algorithm to reanalysis-driven regional climate simulations over Europe (12 km resolution) shows similar performance and the subsequently dynamically downscaled simulations (2 km resolution) well reproduce the observed precipitation statistics of the PEDs from the training period. Additional tests on continuous 12 km resolution historical and future (RCP8.5) climate simulations, downscaled in 2 km resolution time-slices, show the algorithm again reducing the number of days to simulate by over 90 % and performing consistently across climate regimes. The downscaling framework we propose represents a computationally inexpensive means of producing high-resolution climate data, focused on extreme precipitation, at the catchment scale, while still retaining the advantages of convection-permitting dynamical downscaling.

## 1  Introduction

Hydrological modellers and regional decision-makers benefit greatly from high spatial $O(1\ km)$ and temporal resolution climate data to both drive their catchment-scale hydrological models and design regional planning strategies. These high-resolution data are necessary as standard-resolution model data $O(10\text{-}100\ km)$ suffer from many deficiencies, most noticeably both "averaging" and "scale-interaction" effects whereby (i) area averaging over large grid cell areas smooths-out fine-scale detail and (ii) feedbacks from small to large scales are not represented (Volosciuk et al., 2015); these deleterious effects are amplified towards

the tails of the distribution (Volosciuk et al., 2015). Despite their desirability, suitably high-resolution datasets are rarely available, either due to the computational expenses associated with running climate models at such high spatial resolutions or, in the case of observations, due to insufficiently dense observational networks. To bridge this gap, both statistical and dynamical downscaling techniques have been developed for precipitation (Maraun et al., 2010) and other variables.

Statistical downscaling, encompassing a range of approaches (Wilby and Wigley, 1997) in which empirical relationships between large-scales and local weather (i.e. observations) are developed, allows large ensembles of high-resolution climate data to be produced from coarse-resolution models at minimal computational expense and tailored to specific end-user needs. Such relationships can however only be developed in the presence of both appropriate local weather data (typically observations) and corresponding large-scale data (reanalysis or observational data), which are often unavailable at sub-daily and

sub-hourly temporal resolutions and/or spatially too sparse. Dynamical downscaling with regional climate models (RCMs), $O(10$ km), provides an alternative to the statistical approach, which is however computationally far more expensive. Issues of computational expense aside, both methods have their own strengths and (sometimes common) weaknesses. The representation of large scales in the parent general circulation model (GCM) can be a limiting factor, the so-called "garbage in, garbage out" problem (Rummukainen, 2010). If the large scales are not skilfully represented, then downscaling techniques cannot add

value (Benestad et al., 2008) as errors in the large scales will not be corrected; isolated examples of value being added via RCMs correcting large-scale errors have, however, been reported (e.g. Veljovic et al., 2010). Assumption of stationarity – that predictor-predictand relationships will remain unchanged in a future climate – in RCM parametrizations and statistical downscaling methods may also not be valid (Takayabu et al., 2016), lowering confidence in projections. Statistical and dynamical downscaling both produce climate change signals which are, to varying degrees, influenced by the climate change signal of

the parent GCM. If the GCM has an incorrect climate-change signal this may be inherited without meaningful modification. Takayabu et al. (2016) further discuss different facets of the statistical and dynamical downscaling approaches, additionally explaining that the approaches are complementary and can be combined, rather than being treated as mutually exclusive alternatives.

In general, high-resolution RCMs ($\sim$10 km) add value to coarser GCMs for multiple variables (Feser et al., 2011). This

added value (AV) is primarily achieved through better representation of surface forcings and mesoscale processes, and is thus most evident in the presence of complex topography (Heikkilä et al., 2011; Torma et al., 2015) or strong land-sea contrasts (Feser et al., 2011). For example, recent studies have shown cases in which high-resolution RCMs can not only modify but even reverse the mean-precipitation climate-change signal in their parent GCM (Torma et al., 2015), which is attributable to their representation of complex topography and ability to hence simulate increased convective activity at higher elevations in a

warmer climate. Precipitation, due to its high spatial and temporal variability, is perhaps the variable for which high-resolution RCMs exhibit the most AV. The strongest manifestations of AV for precipitation are found at short temporal scales, in the warm season, and in regions of complex topography regardless of temporal scale and season (Di Luca et al., 2012); AV is most evident for the extremes (Heikkilä et al., 2011). Importantly, this AV should not simply be understood as representing increased small-scale detail, but rather AV at the spatial scale of the driving GCM due to more processes being represented (Torma et al.,

2015). As input for impact and hydrological models, dynamical downscaling can provide a large set of physically-consistent

variables (Rummukainen, 2010), meaning that, e.g., changes in cloud cover will be reflected in appropriate knock-on effects on other input variables such as radiation, temperature, humidity, surface pressure, etc.

Despite their relatively high resolution, typical RCMs $O(10 \text{ km})$ still cannot resolve many precipitation-causing processes such as convection, which must instead be parametrized. As a result, models with parametrized convection tend to misrepresent heavy precipitation events, causing them to be too temporally persistent, too spatially widespread and locally not intense enough (Kendon et al., 2012); further issues are too much drizzle (Boberg et al., 2009) and a temporally displaced diurnal convective cycle (Hohenegger et al., 2008). Increasing horizontal resolution below about 4 km, convection-permitting models (CPMs) can explicitly simulate deep-convective processes and improve on many of these shortcomings (Prein et al., 2015). The explicit representation of convective dynamics in CPMs produces more realistic convective features (Weisman et al., 2008), more accurate local precipitation intensities (Lean et al., 2008), and an improved representation of the diurnal convective cycle (Prein et al., 2013). With respect to the accuracy of precipitation totals, the main AV of CPMs can be expected to be found in area averages over, for example, a river catchment (Roberts, 2008). Importantly, the AV of CPMs is not restricted to improved present-climate precipitation statistics (e.g. Ban et al., 2014), but may also extend to the climate change signal. Recent studies show that sub-daily convective extremes in CPMs exhibit an amplified response to enhanced boundary forcings compared to that found in their coarser parametrized-convection parent models (Kendon et al., 2014), which can be highly non-linear (Meredith et al., 2015). The explicit simulation of physical process chains in CPMs, which can be highly-localized, gives more confidence in their projections than those derived from models using convective parametrizations.

CPMs provide a reliable and state-of-the-art means of downscaling coarse-model output to the high spatial-resolutions (with fine-scale variability) needed by hydrologists and end-users for many applications, particularly the study of extremes. A serious limitation of CPMs, however, is the considerable computational expense incurred when carrying out convection-permitting simulations on multi-year timescales, making them an infeasible option for many; an approach for limiting these costs must be sought. For users interested in studying the impact of heavy or extreme precipitation events on their catchment, at least 90 % of the days in any continuous simulation will be of little interest and could be viewed as wasted computational time. In an ideal procedure, dynamically downscaling to convection-permitting resolution might be skipped on these redundant days and only be carried out when there is a significant chance of the catchment experiencing heavy precipitation. Similarly, some users are more interested in assessing the catchment-scale impacts of a selection of physically-plausible extremes from a present or future climate, without being focused on precise probabilities derived from continuous CPM simulations (Hazeleger et al., 2015); examples of this include design situations for hydraulic infrastructure, process-oriented case-studies, and stress testing. The identification of which days to downscale, however, is a non-trivial task. Coarse model precipitation on its own is a poor predictor of extreme precipitation events in both observations and CPMs, especially in the summer, when precipitation extremes tend to be short-duration and of a convective nature (Fig. 1).

With the aim of slashing computational time and expense, we develop a transferable methodology to discriminate between days with an increased likelihood of extreme precipitation – "potential extreme days" (PEDs) – and redundant days so that dynamical downscaling to convection-permitting resolution can be performed over a catchment only when a day has been

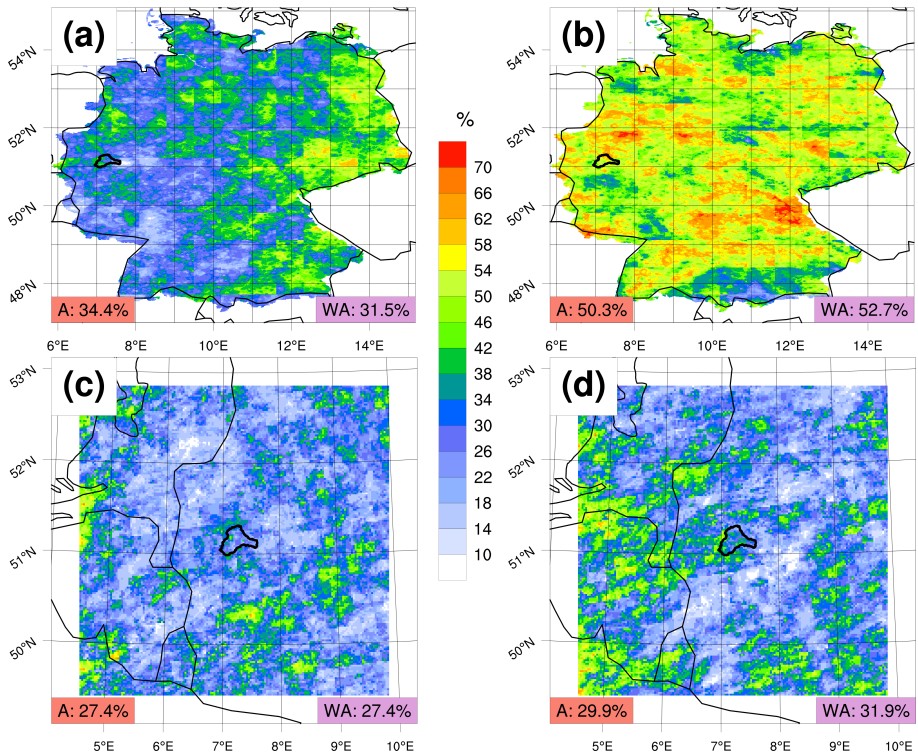

**Figure 1.** Coarse model extreme precipitation is a poor predictor of extreme precipitation in both observations and high-resolution simulations. Plots show the rate at which extreme precipitation events in a coarse model are temporally/spatially coincident with extreme precipitation events in (a, b) observations and (c, d) further downscaled high-resolution simulations. (a) For summer extreme precipitation (1979-2015), the percentage of 99th percentile days in ERA-Interim (Dee et al., 2011) for which the corresponding day in observations (REGNIE; Rauthe et al., 2013) exceeds the *observed* 99th percentile; percentiles are over all days. A value of 100% would mean that, for a given grid cell, all 'extreme' dates in ERA-Interim were also 'extreme' dates in REGNIE. (b) As in (a), except for winter (1980-2015). (c), (d) As in (a), except between the 0.11° and 0.02° CCLM simulations discussed in Sect. 2 for the (c) historical (1970-1999) and (d) RCP8.5 (2070-2099) periods. Values in the bottom-left of each panel show the area average over all data points, while values in the bottom right show area averages over the Wupper catchment in western Germany (marked; see also Sect. 2).

identified as a PED. In Sect. 2 we set out in detail our methodology and validation approach, with the subsequent sections containing results, discussion and conclusions.

## 2 Methodology and Data

To identify for dynamical downscaling days with an increased likelihood of extreme precipitation – "potential extreme days"
5 (PEDs) – over the region of interest, we develop a two-step classification method based on (1) the synoptic-scale circulation pattern and (2) local-scale (modelled) meteorological predictors in the coarser-resolution parent model. This requires the iden-

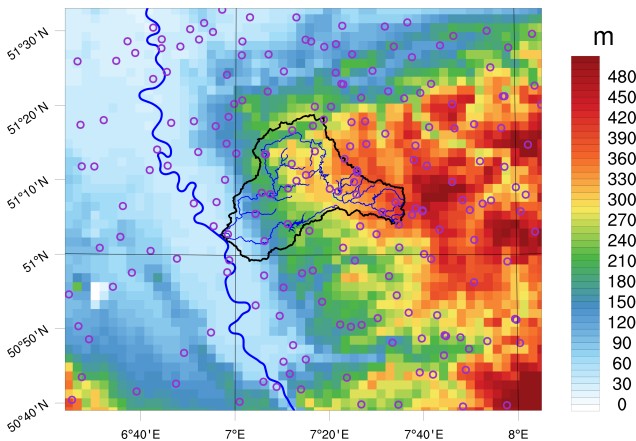

**Figure 2.** The Wupper catchment (black outline) with main tributaries and lakes, and the River Rhine running north-northwestwards. Shading represents the regional orography as represented in the 0.02° CCLM model used in the simulations (see Sect. 2.3). Note that this is is not the full 0.02° simulation domain, but rather a zoom-in over the Wupper catchment; the full spatial extent of the CPM domain and the exact region covered by this map are marked in the inner box of the top-left panels in Figs. 3 and 4. Magenta-coloured circles mark precipitation-recording stations of the German weather service, as listed here <https://www.dwd.de/DE/leistungen/klimadatendeutschland/statliste/statlex_html.html?view=nasPublication&nn=16102.html>. Note that some stations do not cover the entire 1979-2015 period.

tification of synoptic-scale circulation patterns which typically accompany extreme precipitation events in our catchment and the careful selection of meteorological predictors which, when in the vicinity of the catchment a defined threshold is exceeded, are conducive to the development of intense precipitation.

Our study catchment is that of the River Wupper in western Germany (Fig. 2). The Wupper catchment, home to some 5   950,000 inhabitants, has an area of 813 km$^2$, contains about 2,300 km of streams and rivers, and drains into the River Rhine. The Wupper basin is vulnerable to winter flooding and summertime flash-flooding from mesoscale convective events; we thus focus on these two seasons.

## 2.1   Identification of synoptic-scale extremal circulation patterns

The REGNIE gridded daily precipitation dataset (Rauthe et al., 2013), developed by the German weather service specifically 10   for hydrological applications and with a grid spacing of roughly 1 km, is used to compute separate time series of observed daily precipitation area-averaged over the Wupper catchment (Fig. 2) for each full winter and summer in the period 1979-2015. From these time series the 99$^{th}$ precipitation percentiles of all days are computed separately for each season, and all days above their *seasonal* 99$^{th}$ are defined as 'extreme'. The areal extent of the Wupper catchment contains 753 REGNIE grid cells; precipitation-recording stations of the German weather service are marked in Figure 2. An advantage of the REGNIE dataset 15   is that measured totals are conserved, so that observed events (dry or wet) can be found preserved in the gridded field, which is in contrast to other methods on coarser grids which use smoothing (Rauthe et al., 2013). Despite this, the usual warnings about using gridded observations to study heavy precipitation events must be recalled. In the absence of a sufficiently-dense

rain-gauge network in and around the catchment, the spatial variability and local intensity maxima of heavy precipitation events will not be captured in the gridded product, leading to precipitation extremes which are both underestimated and too spatially homogeneous, in particular in areas of complex topography and for convective events (e.g. Hofstra et al., 2010; Ly et al., 2011). The rain-gauge network underlying the gridded dataset must thus be sufficiently dense so that catchment-relevant

extremes are acceptably captured. Alternatively, individual station(s) known to be broadly representative could be used for small- to medium-sized catchments.

To identify the large-scale circulation patterns associated with the heavy rainfall days, the corresponding 500 hPa geopotential height (Z500) anomalies are extracted from the ERA-Interim reanalysis (Dee et al., 2011). REGNIE precipitation has a measurement period of 0730-0730 local time, equating to 0530-0530 UTC in summer and 0630-0630 UTC in winter. Z500

anomalies are thus averaged over the timesteps 12, 18 and 00 UTC, i.e. the middle of the accumulation period, and are relative to their 1979-2015 seasonal means.

The extracted Z500 anomaly patterns next undergo a cluster analysis via the simulated annealing and diversified randomization (SANDRA) method (Philipp et al., 2007). SANDRA has been shown to overcome many of the limitations of standard k-means clustering algorithms, greatly reducing the role of stochastic effects in the final cluster partitions and thus providing

clusters much closer to the "global optimum" (Philipp et al., 2007). It is also less numerically costly than model-based clustering algorithms such as Gaussian mixture models (e.g. Rust et al., 2010). Relevant software for meteorological applications has been developed in the EU COST Action 733 (Philipp et al., 2016), and we use this software in our study. Geopotential height is a standard variable for cluster analyses of atmospheric circulation patterns (e.g. Hidalgo-Muñoz et al., 2011; Merino et al., 2016; Romero et al., 1999). Following Brigode et al. (2013), the spatial extent of the clustering domain is subjectively

chosen such that the typical synoptic patterns associated with extreme precipitation in the Wupper catchment can be captured within the domain when present (Figs. 3-4), which is easily identifiable from historical extremes. Prior to the cluster analyses, outliers which would have little chance of being assigned to an appropriate cluster are removed from the datasets. Outliers are identified by computing, for each day, the Pearson pattern correlation of each Z500 anomaly pattern with that on all other extreme days; any day whose maximum pattern correlation (i.e. across all days) is more than two standard deviations below

the sample mean of the same is excluded from the cluster analysis. In our case, this results in just one day being removed from each of the winter and summer input data, leaving 31 and 33 days respectively. As a stability criterion, the number of clusters $K$ is increased until the minimum intra-cluster pattern correlation – that is, the Z500-anomaly pattern correlation between each cluster member and its own cluster mean – is not less than 0.5. This way all days are assigned to a cluster with which they have genuine similarities, rather than simply the error-minimized 'least bad' cluster, as is typically the case in clustering large

datasets of meteorological variables.

The resulting Z500 anomaly clusters and any outliers are considered as 'reference' extremal circulation patterns against which candidate days from a given dataset can be classified as PEDs, based on their similarity to these references. To this end, the area-weighted Pearson pattern correlation $\rho_{i,j}$ (uncentred) between the Z500 anomaly fields of the candidate day $i$ and the cluster centroid $j$ is used; for our clustering domain (Figs. 3-4) this encompasses 1,935 data points (i.e. grid cells). A perfect

$\rho_{i,j}$ would have a value of 1. With the guiding aim of correctly classifying as many extreme days (i.e. P $\geq$ P$_{99D}$) and rejecting

as many non-extreme days as possible, a $\rho$ threshold ($\rho_{jt}$) is chosen for each cluster centroid $j$ and days with a $\rho_{i,j}$ below this threshold are rejected. $\rho_{jt}$ for each cluster is simply the minimum intra-cluster pattern correlation, reduced by 10 % so that days with a $\rho$ comparable to the lowest intra-cluster $\rho$ are not rejected. To account for clusters with a particularly high $\rho_{jt}$ due to few members, $\rho_{jt}$ is capped at $\frac{2}{3}$.

## 2.2  Assessment of local-scale meteorological predictors

All remaining days not rejected based on their $\rho_{i,j}$ are next assessed in terms of relevant meteorological predictors at the local-scale, i.e. in the vicinity of the catchment. The choice of meteorological predictor and the area around the catchment in which it is assessed are flexible. In general, they may depend on the catchment, season and variables available from the coarser parent model. Chan et al. (2018) advise choosing predictors that are easy to diagnose from coarse-resolution models and consistent with meteorological knowledge of precipitation extremes, e.g. circulation and stability metrics. Guidance may also be sought from statistical downscaling techniques which have been successfully applied in the region. For the Wupper catchment in summer (JJA) and winter (DJF) we select daily maxima (0600-0559 UTC) of relative humidity (700 hPa JJA, 300 hPa DJF) as an indicator of (near-)saturated air masses in the troposphere, 500 hPa horizontal divergence (JJA, DJF) as an indicator of tropospheric vertical ascent (of a frontal or convective nature), convective available potential energy (CAPE; JJA) as an indicator of atmospheric instability, and daily accumulated coarse-model precipitation (JJA, DJF). As with the Z500 data, variables are extracted from ERA-Interim on a Gaussian N128 grid ($\sim$0.7°). To account for the transient nature of many extreme weather systems and the often low temporal resolution of reanalysis/model data (e.g. 6-hourly in the case of ERA-Interim), it is not only the nearest ERA-Interim grid cell to the catchment centre which is considered, but an entire 7x7 cells around it (3x3 in the case of coarse-model precipitation). With the guiding aim of 'catching' the highest number of observed precipitation extremes (i.e. P $\geq$ P$_{99D}$) while excluding as many other days as possible, exceedance thresholds for each variable are *empirically* chosen, either as exceedances of a given percentile (divergence, CAPE, coarse-model precipitation) or as absolute values (relative humidity). The thresholds used for the Wupper catchment are summarized in Table 1. To account for different model climatologies and thus facilitate transferability to other models, the (absolute) relative humidity threshold ($RH_{thresh}$) determined from the training data can be redefined as a function of the model's climatological mean ($\overline{RH}$), i.e. $RH_{thresh} = A \cdot \overline{RH}$, with $A$ a constant; this function can be applied to another model's $\overline{RH}$ to get $RH_{thresh}$ for that model.

In order to be classified as a PED, each threshold must be exceeded at any one of the grid cells (not necessarily the same cell) around the catchment. A schematic summarizes the full two-step selection algorithm (Algorithm 1). Extremal patterns identified for the Wupper catchment are presented in Sect. 3.1.

## 2.3  Validation and simulation

The combination of variables, thresholds and clusters for detecting observed precipitation extremes and excluding non-extreme days is, as discussed above, empirically determined on the basis of the ERA-Interim and REGNIE datasets. Once this has been achieved, the method is applied identically to 0.11° ($\sim$12.2 km) evaluation simulations over the pan-European EURO-CORDEX domain (Jacob et al., 2014), roughly 25-72°N/20°W-50°E, covering the period 1979-2015. Simulations were per-

**Table 1.** Predictor variables, thresholds and region. Note that these thresholds are relative to the model's/reanalysis' own climatology, so that the absolute values of the anomalies/percentiles will vary depending on the model/reanalysis on which the classification algorithm is being applied. On the Gaussian N128 grid, one cell has a width of roughly 75 km. These predictors/thresholds could be used as a starting point if applying the method to other catchments, though should not be directly transferred without first considering meteorological characteristics specific to heavy rainfall events in the new catchment.

| Variable | Threshold DJF | Threshold JJA | Time method | Cells |
|---|---|---|---|---|
| Horizontal divergence (500 hPa) | 90th percentile | 90th percentile | Daily Maximum | 7x7 |
| Relative humidity | 97.5 % (300 hPa) | 86 % (700 hPa) | Daily Maximum | 7x7 |
| CAPE | n/a | 90th percentile | Daily Maximum | 7x7 |
| Model Precipitation | 95th percentile (all days) | 95th percentile (all days) | Daily sum | 3x3 |

formed with the regional climate model COSMO-CLM (CCLM; Rockel et al., 2008) version 4.8, with ERA-Interim reanalysis (Dee et al., 2011) as lateral boundary forcing. CCLM is the community model of the German regional climate research community jointly further developed by the CLM-Community. The years 1989-2008 were simulated by the CLM-Community as part of the EURO-CORDEX experiment (Kotlarski et al., 2014). Years 1979-1988 and 2009-2015 (up to 31.07.2015) were
simulated by the present authors using identical model version and settings.

Z500 CCLM data are interpolated to the clustering domain and the selected meteorological variables are conservatively regridded to a grid of similar spatial resolution to that used in the training stage, i.e. $0.7°$ and centred on the Wupper catchment. All winter and summer days are then either classified as PEDs for further dynamical downscaling with CCLM to a convection-permitting resolution of $0.02°$ ($\sim2.2$ km) or rejected; the nesting ratio of 5.5:1 is in line with that recommended in the literature
(Denis et al., 2003). The enhanced performance of CCLM at convection-permitting resolution (relative to coarser resolutions) in reproducing precipitation statistics, particularly extreme statistics, over central Europe has been extensively documented (Ban et al., 2014; Fosser et al., 2015; Brisson et al., 2016b).

The additional downscaling step is performed using the same version of CCLM with a 221x221 grid cell domain centred on the Wupper catchment (Figs. 3-4), giving sufficient spatial spinup (Brisson et al., 2016a) upstream. 161 of the CCLM grid cells
fit inside the catchment. The simulations are carried out in 'weather forecast mode', i.e. initialized with interpolated values from the parent model. The multi-year simulations of the parent model ensure that soil moisture and temperature are spun-up at the 12 km scale, though not necessarily at the scale of the CPM. The soil moisture climatology tends to be drier in CPMs due to the sparser nature of their precipitation events (Kendon et al., 2017). While studies suggest that the transient boundary conditions are of first order importance for the occurrence of precipitation (e.g. Pan et al., 1999), precipitation extremes highly
sensitive to localized soil-moisture anomalies may be inadequately represented under such a procedure.

**Algorithm 1** Schematic of **classification algorithm** for identifying PEDs in summer. Example for a single day $i$.

$\rho_{i,j}$ is the Pearson pattern correlation between day $i$ and extremal pattern $j$, *RH700* is relative humidity at 700 hPa, *DIV500* is horizontal divergence at 500 hPa, *CAPE* is convective available potential energy, *P* is accumulated daily precipitation.

$\rho_{jt}$ (i.e. $\rho$ thresholds) are determined as described in Sect. 2.1. **if** tests of local-scale meteorological variables are performed using the thresholds and grids described in Table 1. If *any* of the cells in the grid pass the test, then the next test is applied.

For winter the algorithm is the same, except that CAPE is excluded and relative humidity is at 300 hPa.

```
for j in (1,…,K) do                                                    # Extremal patterns 1 to K
    if (ρ_{i,j} ≥ ρ_{jt}) then                                          # Synoptic-scale tests
        if (RH700_i ≥ RH700_thresh) then                                # Local-scale tests
            if (DIV500_i ≥ DIV500_thresh .OR. CAPE_i ≥ CAPE_thresh) then
                if (P_i ≥ P_95D) then
                    DAY_i classified as PED
                end if
            end if
        end if
    end if
end for
```

Lateral boundary conditions are updated 3-hourly and 50 unevenly spaced terrain-following vertical levels are used. For each identified PED, the $0.02°$ simulation is initialized at 1200 UTC the preceding day to allow abundant precipitation spin-up time; as little as 3-6 hours are typically sufficient in convection-permitting models though (Sun et al., 2012). PEDs on consecutive days are downscaled continuously to save resources. For validation, the precipitation statistics of the dynamically downscaled PEDs from the CCLM evaluation runs are compared with those of the observed PEDs identified from ERA-Interim. Area averages of daily precipitation over the Wupper catchment are considered, using REGNIE and $0.02°$ model output. The REGNIE and CCLM grids are of similar spatial resolution (1 km and 2.2 km, respectively). Users should nonetheless be cognizant that datasets of different resolution may exhibit differing statistical characteristics simply because of their different resolutions, e.g. for the area mean. The evaluation and validation of the identified PEDs is presented in Sect. 3.2.

## 2.4 Verification via seasonal time-slice simulations

To provide a sterner test of the method, we additionally perform two sets of 30-season convection-permitting time-slice simulations over the Wupper catchment so that the method can also be assessed in reverse – of the actually simulated 0.02° extreme days ($P \geq P_{99D}$), how many would have been identified as PEDs from the 0.11° coarse model?

A different GCM – the Max Planck Institute's Earth System Model (MPI-ESM-LR) – at the start of the modelling chain provides a new challenge for the method from the previous ERA-Interim-driven simulations. The MPI-ESM-LR runs are continuous transient simulations performed as part of the CMIP5 project (Taylor et al., 2012), using observed greenhouse gas concentrations from 1949-2005 (historical) and representative concentration pathway 8.5 (RCP8.5; Van Vuuren et al., 2011) from 2006-2100. One MPI-ESM-LR member (1949-2100) has been continuously downscaled with CCLM over the EURO-CORDEX domain to 0.11° resolution by the CLM-Community (Keuler et al., 2016); model settings are as in the previously discussed ERA-Interim-driven evaluation runs, greenhouse gas concentrations excepted.

For the present study, we have dynamically downscaled the aforementioned 0.11° CCLM transient simulations to 0.02° over 30 summers from the historical and RCP8.5 periods, 1970-1999 and 2070-2099 respectively. The 0.02° model domain and setup are the same as in Sect. 2.3 (greenhouse gas concentrations aside); simulations are initialized in April and run *continuously* until the end of August each year, with analysis restricted to JJA. Summertime extreme precipitation in the Wupper basin tends to be of a convective and more catchment-scale nature than its wintertime counterpart, with small-scale variability and chaotic processes playing an enhanced role in event intensity. In addition to this, potential differences in large-scale circulation found in a future climate pose an additional challenge for the classification algorithm. The choice of 30 summers, historical and future, is thus intended to ensure a robust testing of our method. The performance testing via seasonal time-slice simulations is presented in Sect. 3.3.

## 3 Results and Discussion

### 3.1 Extremal circulation patterns for the Wupper catchment

The greater diversity of synoptic patterns which can lead to extreme precipitation in the Wupper catchment in summer, compared to winter, can be seen in the number of clusters $K$ necessary before our stability criterion (see Sect. 2.1) is reached (Figs. 3-4). The higher $K$ also hints at the in general more challenging nature of forecasting summertime intense precipitation, when synoptic forcing tends to be weaker and small-scale chaotic processes play an increased role. In winter (Fig. 3), precipitation extremes in the Wupper catchment are most often associated with a dipole-like structure characteristic of a strong positive phase of the North Atlantic Oscillation (Hurrell, 1995), with various shifts of the dipole centres (clusters 1-3). Such a synoptic pattern gives a strong zonal forcing, sweeping deep low-pressure systems and associated frontal precipitation across the catchment; similar clusters have previously been identified for north-eastern Switzerland (Giannakaki and Martius, 2016). For the remaining cluster (cluster 4) and the outlier, shallower depressions become embedded in a relatively weak zonal flow, depositing their albeit less intense precipitation over a more prolonged period; these patterns account for less than one sixth of all extreme days

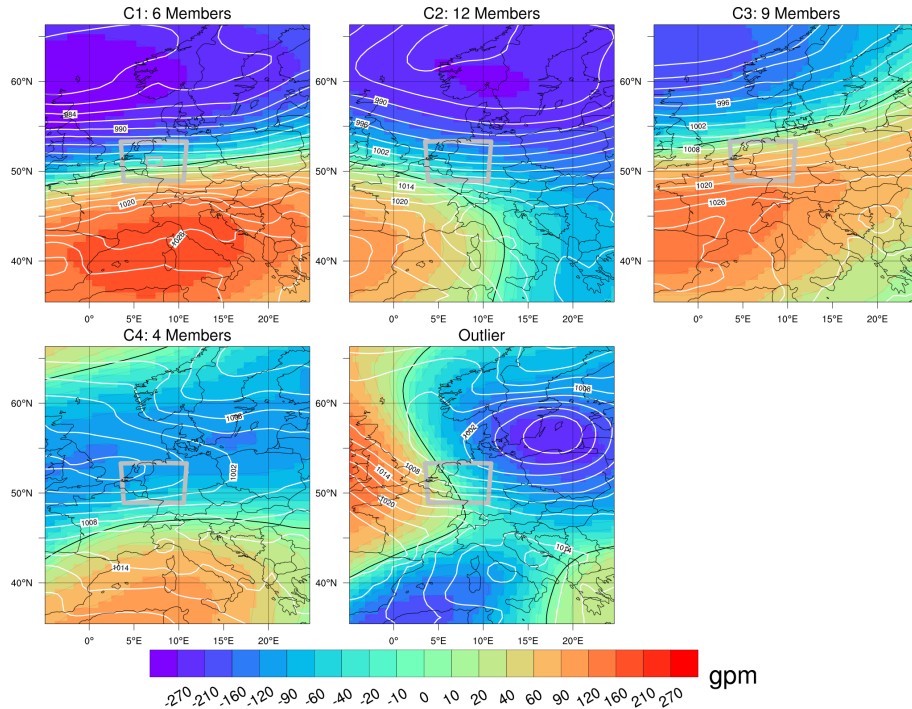

**Figure 3.** 500 hPa geopotential height anomalies (shading) of extremal circulation patterns identified for the Wupper catchment in winter, via the clustering algorithm, and one outlier; the zero-line is marked in black. White contours represent the accompanying sea level pressure patterns. The grey box centred over western Germany is the 0.02° simulation domain (Sect. 2.3).

($P \geq P_{99D}$) though. In summer, a dipole-like pattern can also be seen on some extreme days (cluster 1), though accounting for just over one seventh of all extremes; such events in summer can also be expected to include enhanced frontal convection. The remainder of the summertime extremes are associated with a weaker zonal forcing. High pressure over eastern Europe often advects warm, moist air from the Mediterranean into central Europe (clusters 2 and 4), enhancing instability and increasing
5  the chance of deep convection; Bárdossy (2010) also identified such a pattern as bringing intense precipitation to south-west Germany during summer. Another common pattern is that of a front, often with a small embedded low, extending across the catchment (clusters 3 and 8) in the wake an eastward moving ridge and triggering frontal lifting as it passes. Quasi-stationary mid-tropospheric cut-off lows (clusters 5-7) are the most common cause of summertime extremes in our catchment, allowing slow-moving surface lows to advect a persistent moisture stream, within which intense convective cells can develop, across
10  the catchment. A not dissimilar pattern was also identified by Brigode et al. (2013) in their study of extreme precipitation in Austria.

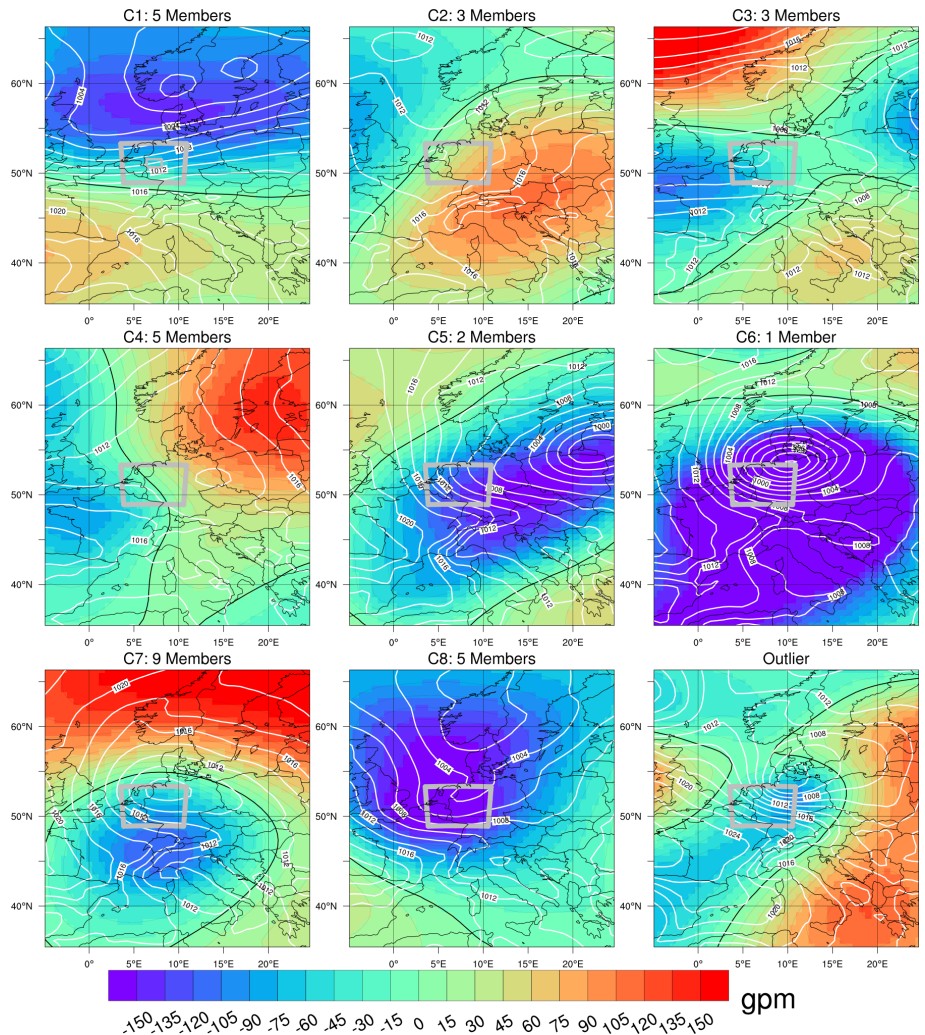

**Figure 4.** As in Figure 3, except for summer.

## 3.2 Evaluation and validation of identified PEDs

While still capturing more-or-less all observed extreme days, the constraints derived from ERA-Interim variables enable the classification algorithm to reduce the number of PEDs to well below 10 % of all days (Table 2). In the process, the number of "redundant days" (i.e. $P < P_{90D}$) falls from about 3,000 to 48 in winter and 126 in summer. The "redundant days" thus occupy a much smaller fraction in the PEDs than in the set of all days. Such a good performance in the training dataset is, however, no surprise.

Applying the same methodology to the 0.11° CCLM evaluation runs (ERA-Interim driven) over the same period, a similar number of PEDs are identified for dynamical downscaling to 0.02° (Table 2). The PEDs again represent well below 10 % of

**Table 2.** Summary table of performance of classification algorithm for training period (ERA-Interim) and CCLM evaluation runs. "*Redundant days*" are defined as days with precipitation below the 90[th] percentile (all days). The third column shows the percentage of total days identified as PEDs, with the fourth column showing the percentage of actual extreme days contained within these PEDs. The rightmost column compares the fraction of redundant days ($P < P_{90D}$) contained in the PEDs and amongst the set containing all days ("All Days").

| Data / Experiment | Time Period (# days) | PEDs (# days) | $P_{99D}$ captured (days/total days) | Redundant days (days/total days) | |
|---|---|---|---|---|---|
| | | | | **PEDs** | **All Days** |
| ERA-Interim | **DJF** 1980-2015 (3,249) | 6.8 % (222) | 100 % (32/32) | 22.5 % (50/222) | 90.0 % (2,924/3,249) |
| ERA-Interim | **JJA** 1979-2015 (3,373)[†] | 8.6 % (290) | 97 % (33/34) | 44.1 % (128/290) | 90.0 % (3,036/3,373) |
| CCLM-0.11° CORDEX-EU (ERA-Interim driven) | **DJF** 1980-2015 (3,249) | 6.8 % (220) | n/a | n/a | |
| CCLM-0.11° CORDEX-EU (ERA-Interim driven) | **JJA** 1979-2015 (3,373)[†] | 9.8 % (331) | n/a | n/a | |

†Ends on 31.07.2015.

all days, slashing the computational expense against a continuous simulation of the whole period by an order of magnitude. Of note is that although the 0.11° CCLM simulations are forced at the lateral boundaries by ERA-Interim, only 123 of the 320 dates identified as PEDs in CCLM in summer are also found amongst the ERA-Interim PEDs. This is attributable to the fact that RCMs without interior constraints (i.e. some form of internal nudging) cannot synchronously reproduce the local-scale day-to-day variability of their parent model (Maraun and Widmann, 2015). RCMs of sufficiently large domain size thus often generate large internal variability (e.g. Lucas-Picher et al., 2008), often comparable to that found in GCMs (Christensen et al., 2001), which can cause the local RCM solution to significantly deviate from that of its parent model. In the presence of a strong zonal throughflow, e.g. in winter, the growth of differing internal solutions is limited due to small-scale perturbations being more rapidly swept out of the domain (Giorgi and Bi, 2000). This explains the higher fraction of common days which we find in winter (150/220).

Comparing the empirical cumulative distribution functions (ECDFs) for catchment-averaged precipitation (observed) of all days and PEDs from the training data set (ERA-Interim), the greatly increased probability of heavy precipitation on a randomly selected PED becomes apparent (Fig. 5, blue curve): in the set of PEDs, the probability of exceeding the climatological winter (left panel) 90[th]/99[th] percentile is about 80 %/20 %, whereas in the set of all days it is only 10 %/1 %. For summer (right panel), the situation is less pronounced but the climatological (JJA) 90[th]/99[th] percentile is exceeded on about 60 %/15 % of the days in the PED set. Taking all days, the ECDF can be well described by a typical gamma distribution; the gamma distribution is known to well represent the bulk of the daily precipitation distribution, though perform less well for the tails (Rust et al., 2013). The form of the ECDF of the observed PEDs (blue curve), however, is far removed from that of the set of all days (red

curve), as the probability is shifted towards more intense precipitation. The change in form of the ECDF – from one dominated by dry to light-rain days, to one dominated by heavy- to extreme-rain days – results from the classification algorithm's removal of days with a low potential for intense precipitation.

Dynamically downscaling all CCLM 0.11° PEDs to 0.02° produces ECDFs of daily precipitation which closely resemble those of the observed PEDs, for both seasons (Fig. 5, green curve); both ECDFs are again dominated by heavy to extreme precipitation events, with dry days ($P_D < 0.1$ mm) almost completely eliminated. Indeed, many of the seemingly dry to light-rain days counted over the Wupper catchment in the convection-permitting simulations do still feature heavy precipitation, though displaced to neighbouring regions of the 0.02° simulation domain (Fig. 6); this occurs most often in summer, owing to the more small-scale and chaotic nature of convective precipitation. The good match between the ECDFs of observed and downscaled PEDs shows that, with skilful classification of the PEDs, selective downscaling can be relied on to realistically reproduce the same range of precipitation events over the catchment as expected from the training dataset and observations, allowing of course for known model biases (e.g. Fosser et al., 2015). In the process, computational expense is reduced by over 90 % (Table 2) as compared to the computational efforts which would be required for a continuous simulation over the same period at such high spatial resolution. While the spatial resolutions of REGNIE and CCLM are similar (1 km and 2.2 km, respectively), users should beware that area means in datasets with considerably different grid resolutions may differ simply because of the different sample sizes, i.e. the number of grid cells contained within the averaging area, in particular for small catchments and large differences in grid-box area.

## 3.3   Performance testing on seasonal time-slice simulations

The dynamical-downscaling of two sets of 30-summer time-slices – historical (1970-1999) and RCP8.5 (2070-2099) – from 0.11° to 0.02° provides an additional set of tests for the classification algorithm, namely: what fraction of the actually simulated extreme days in the 0.02° model would the method have identified as PEDs from the 0.11° model? In addition, is classification performance degraded in a future climate? The summer season is chosen to ask these questions due to the greater challenges in predicting summertime intense precipitation (see Sect. 2.4, Sect. 3.1).

Applying the classification algorithm, identically as in Sect. 3.2, to the 0.11° historical and RCP8.5 simulations again yields selections of PEDs representing less than 10 % of the total days (Table 3). Amongst these PEDs, at least 75 % of 0.02°-simulation extreme days are captured in both time-slices. In the case of the historical simulations, the fraction of redundant days (i.e. $P < P_{90D}$) climbs by almost six percentage points relative to the training data set; for the RCP8.5 simulations, the fraction falls marginally. The former increase may simply be an artefact of the fewer summers (30 vs. 37) present in this testing data set. The similarity of performance between the historical and future simulations is noteworthy, particularly in light of RCP8.5 2070-2099 representing the end of the most extreme RCP scenario. Projected changes in large-scale extratropical circulation can be considerable (e.g. Barnes and Polvani, 2013; Zappa et al., 2013), and are known to exert strong control on regional precipitation climatologies (Shepherd, 2014). In the case of the MPI-ESM-LR model used in this study, for example, a northward shift of the annual-mean jet in the Atlantic sector (Barnes and Polvani, 2013) and reduction in the frequency of both North Atlantic and Eurasian summertime anticyclonic blocking (Masato et al., 2013) are projected under the RCP8.5 scenario. Despite this,

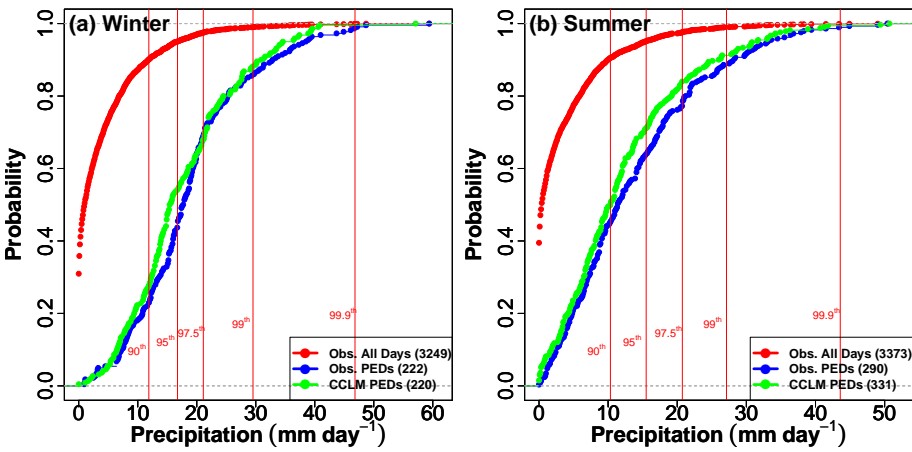

**Figure 5.** Empirical cumulative distribution functions of daily precipitation for all days (red, observed), PEDs (blue, observed), and CCLM PEDs (green, downscaled to 0.02°). (a) Winter 1980-2015, (b) summer 1979-2015 (up to 31.07.2015). Differences between the blue and red curves (REGNIE) highlight the increased likelihood of heavy rainfall events amongst the PEDs. All values are area averages over the Wupper catchment. Vertical red lines mark important percentiles of the all-day distribution. The area of the Wupper catchment encompasses 753 and 161 grid cells of REGNIE and CCLM data, respectively. Stations in and around the Wupper catchment are marked in Fig. 2. The similarity of the blue (REGNIE) and green (CCLM) PED-curves shows that, with skilful identification of PEDs, convection-permitting downscaling can well-reproduce the observed PED statistics.

the classification algorithm performs more-or-less the same in historical and future climates. While the classification algorithm unsurprisingly fails to capture all extreme days in either the historical or RCP8.5 simulations, the fact that the performance is the same across both climates indicates the added value of employing our downscaling methodology, allowing more robust conclusions to be drawn from the output. Of the extreme days which are not captured, 6 out of 7 (historical) and 4 out of 5

5   (RCP8.5) are lost due to their circulation patterns not well matching any of the pre-defined extremal clusters. This could indicate that the training period for identifying the extremal patterns is too short to encompass sufficient diversity or, more likely, that the GCM in question (MPI-ESM-LR) does not adequately represent the frequency and/or persistence of the large-scale circulation patterns which lead to observed extremes in our catchment. For example, CMIP5 GCMs are known to underestimate the frequency of Eurasian blocking (Masato et al., 2013) and GCMs in general often underestimate the persistence of blocking

10   systems (e.g. Matsueda, 2011); the poleward flank of such blocking anticyclones often transports warm moist air into central Europe enabling intense convective precipitation (see Sect. 3.1). In the case of MPI-ESM-LR during summer, a southward bias in the storm-track axis and over-active North Atlantic blocking are also evident in the CMIP5 historical simulations (Masato et al., 2013).

    The similar performance of the classification algorithm across climates, as well as the evaluation period, is confirmed by

15   looking at the historical and RCP8.5 ECDFs (Fig. 7). As in the training dataset, the ECDFs of the PEDs are shifted towards more intense precipitation compared to the ECDFs for the sets of all days. For the PEDs, the probability of exceeding the respective climatological (JJA) 90[th]/99[th] percentile in the historical and RCP8.5 simulations is similar to that found in the

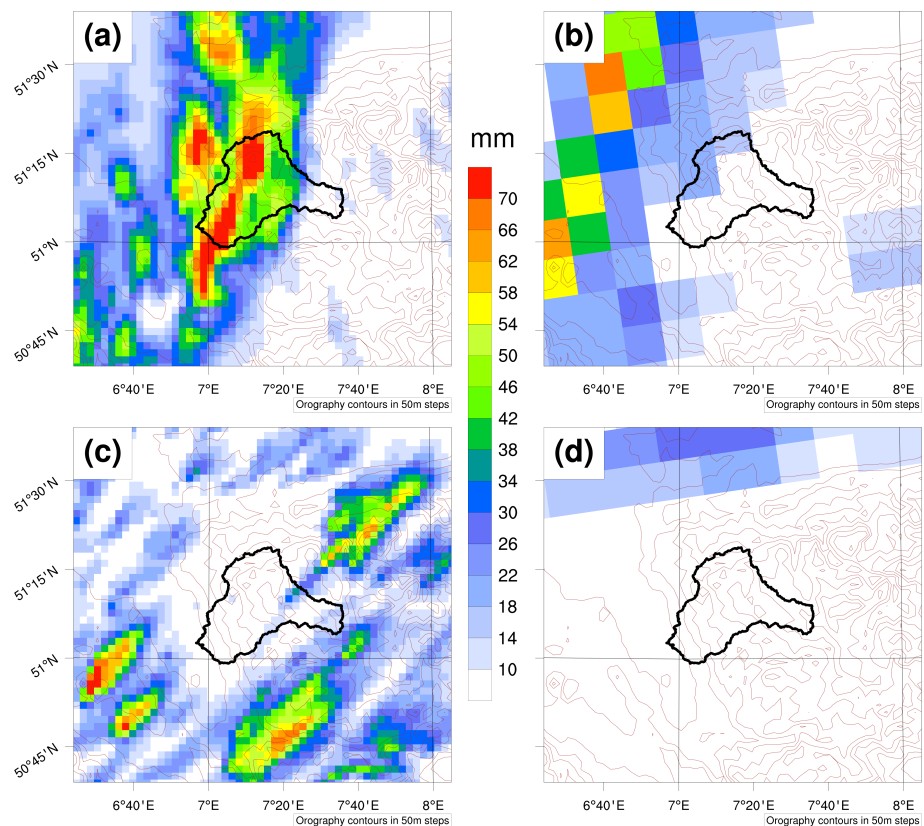

**Figure 6.** Illustrative modelled PEDs. (a) Example summer PED downscaled to $0.02°$ and (b) the same day in the $0.11°$ parent model. In this example, the strongest precipitation directly strikes the catchment in the $0.02°$ CCLM despite missing the catchment in the parent $0.11°$ CCLM. (c) Example summer PED with highly localised intense precipitation which falls outside the catchment in the $0.02°$ CCLM. (d) The corresponding day in the $0.11°$ CCLM.

training dataset and the dynamically downscaled PEDs of the evaluation period, roughly 55 %/10 % (as compared to 10 %/1 % for all days), and the ECDFs are dominated by heavy to extreme events with dry days almost absent. To quantify differences in the distributions of precipitation events amongst all days and the PEDs for discrete intensity ranges, we compute the relative likelihoods ($R$) of finding a precipitation event within a given intensity range for the historical and RCP8.5 simulations (Fig.

5   8); this is simply the ratio of the respective probabilities, e.g. $P(E|PED) : P(E)$, with the smaller of the two probabilities used as the denominator for plotting purposes.

The greatest difference between all days and the PEDs is found in the relative likelihoods of a randomly sampled day being dry, which is an order-of-magnitude lower in the PEDs. Indeed, considering the set of non-PEDs, the probability density function exhibits an even higher density of dry days than found for all days (not shown). Focusing on just wet-day percentiles,

10   a regime shift from a higher $R$ for all days to a higher $R$ for PEDs occurs above the median wet day event. The higher $R$ for the PEDs grows further as event intensity nears the most extreme precipitation events, consistent across historical and RCP8.5

**Table 3.** Summary table of performance of classification algorithm for 0.11° CCLM historical and RCP8.5 simulations, continuously down-scaled to 0.02° over 30 summers. "*Redundant days*" are defined as days with precipitation below the 90[th] percentile (all days). The third column shows the percentage of total days identified as PEDs, with the fourth column showing the percentage of actual extreme days contained within these PEDs. The rightmost column compares the fraction of redundant days ($P < P_{90D}$) contained in the PEDs and amongst the set containing all days ("All Days").

| Data / Experiment | Time Period (# days) | PEDs (# days) | $P_{99D}$ captured (days/total days) | Redundant days (days/total days) | |
|---|---|---|---|---|---|
| | | | | **PEDs** | **All Days** |
| MPI-ESM-LR/CCLM-0.11° CORDEX-EU/*Historical* | **JJA** 1970-1999 (2,760) | 9.8 % (271) | 75 % (21/28) | 49.8 % (135/271) | 90.0 % (2,484/2,760) |
| MPI-ESM-LR/CCLM-0.11° CORDEX-EU/*RCP8.5* | **JJA** 2070-2099 (2,760) | 9.5 % (261) | 82 % (23/28) | 42.9 % (112/261) | 90.0 % (2,484/2,760) |

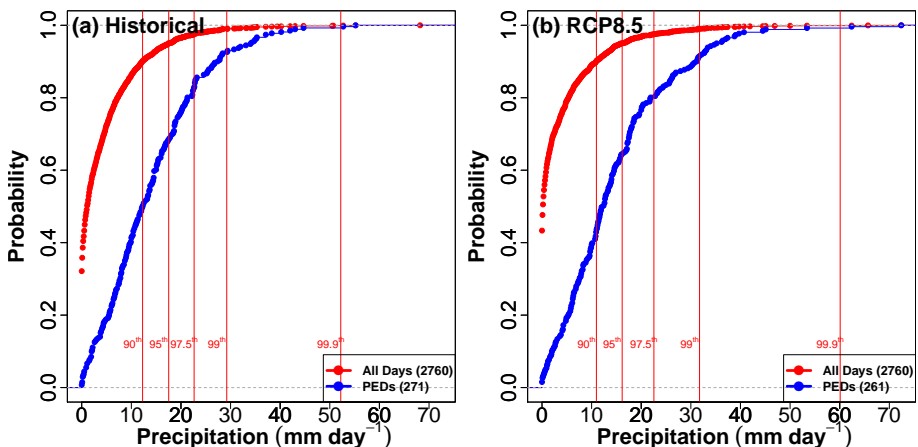

**Figure 7.** Empirical cumulative distribution functions of daily precipitation for all days (red) and PEDs (blue) downscaled to 0.02°. (a) Historical (JJA, 1970-1999), (b) RCP8.5 (JJA, 2070-2099). All values are area averages over the Wupper catchment. Vertical red lines mark important percentiles of the all-day distribution.

experiments and approaching a factor of 10 in places (Fig. 8). For the most extreme events ($P_D \geq P_{W99.9}$), more variability between historical and RCP8.5 $R$-values emerges as the number of days involved limits towards zero. Future changes in the fraction of wet-days, and the sensitivity of wet-day percentiles to such changes (Schär et al., 2016), likely contributes to some of the small differences in relative likelihood between the historical and RCP8.5 experiments.

5  ### 3.4  Applications and Outlook

The preconditioning of PEDs on known extremal circulation patterns does not just reduce the total number of days to dynamically downscale. Importantly, it also allows conclusions to be drawn about changes in catchment-relevant precipitation

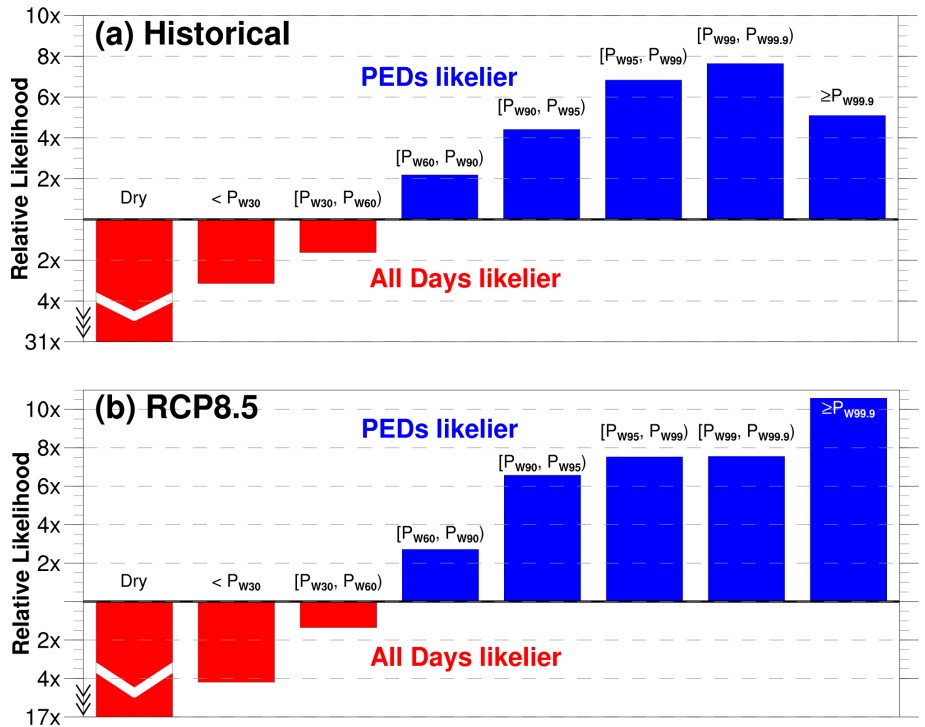

**Figure 8.** Relative likelihoods of precipitation on a randomly sampled day from the set of all days and the PEDs being within a given intensity range for the (a) historical and (b) RCP8.5 0.02° simulations. Note that precipitation intensities are based on the percentiles of wet days ($P_D \geq 0.1$ mm)

between two periods, e.g. present and future climates, for these circulation patterns. A classification method which does not *guarantee* the capture of all extreme days clearly cannot be used to draw overall conclusions about precipitation changes in a given catchment. Preconditioning on circulation types does, however, allow conclusions to be drawn about changes in specific classes of precipitation extreme (Fig. 9), e.g. as identified via the clustering methodology outlined in Sect. 2.1. For example, for

5  a known extremal circulation pattern, will the likelihood that the accompanying precipitation exceeds some catchment-relevant threshold be higher or lower in the future? This approach is in a way analogous to the framework used in conditional event attribution (e.g. Trenberth et al., 2015; Pall et al., 2017), where the question is posed: for some observed circulation pattern, how is the event's intensity affected by known thermodynamic changes in the earth's climate system? A major advantage of such an approach is the relative robustness of projected thermodynamic changes in the climate system compared to projected

10  dynamical changes (Shepherd, 2016). From a catchment-hydrology perspective, one could imagine this being particularly useful for catchments vulnerable to specific compound extremes, for example intense precipitation in an estuarine catchment compounded by a shoreward moving low-pressure system with strong onshore winds (e.g. Bevacqua et al., 2017). Beyond the extremal patterns identified from the training period, however, there remains the possibility that a future climate may also

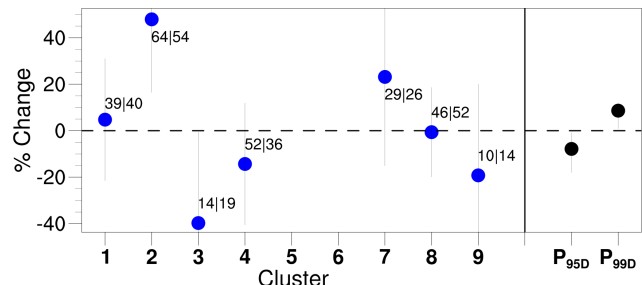

**Figure 9.** Percentage change in daily precipitation intensity between the historical and RCP8.5 periods (JJA), conditional on extremal circulation pattern, from the $0.02°$ simulations. The numbers indicate the total number of PEDs for each pattern (i.e. cluster) in the historical (left) and RCP8.5 (right) periods, while vertical bars represent 90% confidence intervals. Clusters with less than 10 days in either period are excluded from the calculations. On the right hand side, the corresponding climate change signal for the $95^{th}$ and $99^{th}$ percentile of all days is provided for reference.

contain new extremal circulation patterns which were previously either not associated with extreme precipitation or simply not present at all. Such systematic effects could only be explored with continuous dynamical downscaling of the different climates.

## 4   Further Discussion

The consistent performance of the classification scheme across historical and future climates further demonstrates its utility
for studying changes in defined classes of precipitation extreme, for example those preconditioned on an identified extremal synoptic pattern which is known to severely affect a given catchment. In this regard, our method is complementary to current trends in how the projected impacts of climate change are communicated and adapted to end-user needs. Recent literature advocates the use of high-resolution weather models to create bespoke storylines of high-impact weather events for a given catchment in a future climate (Hazeleger et al., 2015). In the so-called 'Tales' approach of Hazeleger et al. (2015), the broad
statistical terms in which climate change projections are typically communicated are replaced by high-resolution simulations of carefully selected past and future weather events over a given catchment in order to study the catchment-specific impacts of individual hydrometeorological events from past/future climates. This approach is designed to form part of a collaborative process in which end-users play a key role in selecting the type of events to be studied, providing vivid case-studies on which adaptation strategies can be decided (Hazeleger et al., 2015). Our methodology could be integrated seamlessly into this
workflow. An additional advantage of this type of modelling framework is that anthropogenic factors extraneous to global climate change can easily be implemented into the modelling chain (Shepherd, 2016), for example adding potential changes in land-use to a high-resolution hydrological model, or changes in hydraulic infrastructure to a hydraulic model for assessing impacts on reservoirs, water-treatment plants, drainage systems, etc.

An important element in the transferability of the method to other catchments is its inherent flexibility, allowing in particular
for an active involvement of end-users. End-users can contribute and integrate their empirical knowledge towards the identi-

fication of the local-scale meteorological predictors most suitable for their own catchment, perhaps taking the ones we use or those suggested in Chan et al. (2018) as a starting point. Data availability in the models to be downscaled may, however, require choosing parameters that are only approximate indicators of the most suitable ones. For the Wupper catchment studied here, for example, we found daily maximum 700 hPa vertical velocity to perform better than daily maximum 500 hPa horizontal

divergence as an indicator of extreme precipitation in the training dataset. The regional model which we wished to downscale, however, had saved vertical velocity at too low a temporal resolution to meaningfully calculate daily maxima. Adoption of horizontal divergence was thus necessitated, allowing the PEDs to still be appropriately classified while avoiding an unacceptable increase in computational expense. The method is additionally adaptable to the computing capacity of the user. With the caveat that excessively high thresholds will result in more undesirably-rejected days, thresholds for the identification of PEDs can be

either raised or lowered based on available computational resources.

Data produced via a method like ours are indeed useful for many applications, though not universally so and do also come with their own limitations. Care must therefore be taken when applying such data and interpreting subsequent results. The issue of stationarity should be acknowledged: one can never be certain that a future climate will not include heavy precipitation events caused by previously unimportant circulation patterns. Non-stationarity may also, positively or negatively, impact

the effectiveness of local-scale predictors. Non-stationarity is indeed a common issue also affecting model parametrization schemes and statistical downscaling – a motivating factor for anchoring our method with a convection-permitting model. Additionally, the catalogue of downscaled PEDs is no random sample of high-resolution climate data and thus cannot be treated as traditional projections. Traditional projections can only be made with continuous, multi-decadal downscaling, and not with the discontinuous time series which we produce.

Our method is instead ideal for applications requiring high-resolution data suitable as input for modelling the catchment-scale impacts of extremes. Such applications include (i) design situations and stress testing for hydraulic infrastructure, e.g. dams, canal networks, urban sewerage systems, and (ii) process-oriented case studies of the catchment's response to extremes, e.g. runoff formation processes leading to flooding. In such applications, the emphasis is on combining realistic initial conditions with physically-plausible and realistic extremes, as input for the hydrological models. Typical problems with using

observational data for such studies are that the spatial and/or temporal coverage of the observational network was insufficient to capture suitably extreme historical events to use in, e.g., design situations. Coarser model data present problems too, in that they also tend not to realistically capture the peak intensities and spatial variability of such intense events (see Introduction). For such studies, hydrological models would need to be calibrated with either observations or lower-resolution RCM data. Realistic initial conditions, e.g. for design situations, can also be obtained from such sources or, as is often the case, prescribed

and varied in order to test the sensitivity to initial conditions of the catchment's response to a given extreme. For example, the reservoir level prior to a rain-on-snow event – such events can quickly mobilise large volumes of water into runoff, potentially overwhelming hydraulic infrastructure.

A further means through which our methodology can be used to limit computational expense is in the selection of individual models from multi-model ensembles (e.g. CMIP) to downscale over a given region, avoiding the computational expense of

dynamically downscaling an entire multi-model ensemble. GCMs whose historical runs fail to satisfactorily reproduce the

observed PED climatology, i.e. the seasonal frequency of PEDs, could be considered to poorly represent the regional extremal circulation patterns and thus be rejected in favour of the top $N$ best-performing models, with $N$ a function of both available computing resources and the reduction in intra-ensemble spread which can be tolerated. Such a region-targeted selection of GCMs (Maraun et al., 2017) could even be combined with the aforementioned 'Tales' approach, making a potent tool.

## 5  Conclusions

Hydrological modellers, amongst others, benefit greatly from high-resolution climate data at the catchment scale – particularly for studying the impacts of extreme precipitation. In achieving these high resolutions $O(1\,km)$ while maintaining data quality, dynamical downscaling to convection-permitting resolution presents numerous advantages, though comes at an often prohibitive computational expense. To reduce the overall computational burden and instead dynamically downscale only those days for which there is an elevated likelihood of extreme precipitation in a catchment, we have developed a flexible and transferable classification algorithm for identifying potential extreme days (PEDs) and rejecting days unlikely to produce intense precipitation. While reducing computational expense by over 90 %, the precipitation distribution of the training dataset's PEDs – in which more-or-less all extreme days were captured – was well reproduced via convection-permitting dynamical downscaling, showing an ECDF dominated by heavy precipitation events. Testing the classification algorithm on continuous datasets driven by a different global model, at least three quarters of the convection-permitting model's summertime extremes – which are intrinsically more challenging to identify than their wintertime counterparts – were caught and computational expenses were again slashed by over 90 %.

Our method represents a computationally inexpensive procedure to produce high-resolution climate data, focused on extreme rainfall events, for hydrological modellers and decision-makers, while retaining the advantages of the convection-permitting modelling framework (see Introduction). The explicit simulation of fine-scale processes along the modelling chain gives additional confidence in the end product, as fine-scale processes can substantially modulate the regional climate change signal (Diffenbaugh et al., 2005). Irrespective of increases in processor power, regional models will always be able to be run at higher spatial resolutions than their global counterparts. Should global models some day run at convection-permitting resolution as standard, classification algorithms can still be utilised for downscaling to ever-higher resolutions at which even more processes can be explicitly simulated, e.g. turbulence. Classification algorithms, such as the one presented here, for selective dynamical downscaling preconditioned on known extremal circulation patterns can thus play an important and enduring role in climate modelling.

*Code and data availability.*  ERA-Interim reanalysis (Dee et al., 2011) are available from the ECMWF (http://apps.ecmwf.int/datasets/data/interim-full-daily). REGNIE precipitation data (Rauthe et al., 2013) are available from the German weather service (DWD, https://www.dwd.de/DE/leistungen/regnie/regnie.html). The 0.11° CORDEX data used within this study are distributed within the CORDEX framework by the Earth System Grid Federation (e.g. https://esgf-data.dkrz.de/projects/esgf-dkrz/). All remaining simulation data and scripts are available from the corresponding author on request.

*Author contributions.* EM developed the method, performed the analysis and wrote the manuscript. HR and UU contributed ideas and comments on the method, analysis and manuscript.

*Competing interests.* The authors declare that they have no competing interests.

*Acknowledgements.* We thank R Benestad and P Laux for their helpful reviews. This study was funded by the European Commission through the H2020 project BINGO (http://www.projectbingo.eu/), Grant Agreement 641739. Simulations were carried out at the North-German Supercomputing Alliance (HLRN) and the German Climate Computing Centre (DKRZ). We thank the German weather service (DWD) for producing and making available the REGNIE precipitation dataset. We thank the EU COST Action 733 for producing and making available the clustering software (http://cost733.geo.uni-augsburg.de). Analyses and plotting were performed with NCL (Version 6.4.0, doi:10.5065/D6WD3XH5) and R. We thank T. aus der Beek, M. Göber, K.A. Kpogo-Nuwoklo, T. Pardowitz, M. Scheibel, C. Vagenas and C. Volosciuk for helpful discussions.

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
