# Peer review of "A classification algorithm for selective dynamical downscaling of precipitation extremes"

_Hydrology and Earth System Sciences, 2017_

## Referee Comment (RC1) · P. Laux (Referee) · 8 Jan 2018

The manuscript presents a very interesting contribution to combine dynamical downscaling approaches with a statistical classification procedure in order to save computational costs. The approach aims at extreme precipitation events and is restricting the dynamical downscaling to those days, in which the probability of extreme events is enhanced. For this reason, the concept of Potential Extreme Days (PEDs) is introduced, which is based on a classification approach of synoptic circulation patterns. The manuscript is well written and understandable in general. The procedure is scientifically sound and clearly described. However, there are concerns in terms of its

"applicability" and "usefulness". In order to deserve publication, the following aspects need to be considered and elaborated.

My main points center around the efforts required to restrict the dynamical downscaling (in convection-permitting resolution) to selected events only and the credibility of the results obtained:

- First, I do not see clearly a potential application behind (at least it is not clearly described in the manuscript). Please elaborate clearly which kind of research and practical application can be considered with this in hydrological modelling.

- In addition, it might be difficult for hydrological models to deal with non-continuous data (time series) focusing on the extreme events only. In particular, issues may arise in calibration/validation of such process-based hydrological models based on extreme precipitation events only, i.e. the credibility might be limited if these models are calibrated based on extremes exclusively.

- The efforts of the classification to identify the PEDs are high. The results depend on the selected domain, number of clusters, selected predictors, selected threshold values, etc. It seems that this is not as straightforward and to be implemented as described in the manuscript. For instance, a predictor screening must be undertaken if the approach is transferred to other regions. Please elaborate and discuss further.

- From regional climate modelling perspectives, I have concerns in selecting single days only instead of performing continuous simulations. I am referring to the initial conditions, when a new simulation is initiated. It is well-known that these are rather imperfect. This is less problematic for the atmospheric compartment of the RCMs (because of the relatively short memory), however, the terrestrial compartments such as e.g. soil moisture need a certain time to reach equilibrium. For this reason, spin-up periods of several days to weeks might be necessary, which limits the benefit of the presented approach tremendously. In addition to that, time requirements to set-up and submit and control multiple short-term simulations are high.

- The application of the classification for the past is well justified, however, it might be very limited for the future ("stationarity" assumption). As correctly mentioned, it can be expected that certain extremal circulation patterns change or other patterns might become more important for extreme events. This is more likely for periods in the far future, e.g. the time slice towards the end of this century, as used in this study. For periods in the far future, I would trust more to the pure dynamical downscaling.

Another concern is the validation of the identified PEDs (section 3.2). I would suggest to check not only the occurrence frequencies, but also the timing of the extremes using the reanalyses data. These can be checked with the timing of the extremes based on station data for the catchment. The frequency is not a good performance criterion to my opinion.

Minor issues:

- I suggest to leave out the code fraction (page 8)

- Section 3.3 (Page 13): The authors claim that they perform a performance testing on continuous simulations, but the tests are restricted to the summer periods. I also understood that the RCM downscaling is done only for the summer periods, but maybe I misunderstood this. Anyway, I think it is confusing and the term "continuos" should be omitted.

- Please check the brackets given after high-resolution data (abstract, line 1, introduction, lines 21 and 23; Page 18, line 2, etc.)

---

## Referee Comment (RC2) · R.E. Benestad (Referee) · 12 Jan 2018

The paper "A classification algorithm for selective dynamical downscaling of precipitation extremes" by Meredith et. al. presents an interesting strategy for a more efficient and targeted simulations of heavy precipitation with high-resolution convective-permitting regional climate models. They make use of the dependency of local rainfall on the large-scale (synoptic) conditions in terms of circulation patterns, and apply a cluster analysis to distinguish between days when the conditions are right for heavy rainfall and days when heavy rainfall is unlikely. Then they run a high-resolution regional climate model based on the first principles (physics-based) to simulate rainfall

for the selected subset. This approach can in a sense be considered as a hybrid between traditional empirical-statistical downscaling and dynamical downscaling, since statistical techniques (clustering) were used to select times for simulations.

The analysis presented in Meredith et. al. are in my opinion scientifically sound and this paper merits publication, but there are a number of important caveats and there are a number of statements with which I think are wrong. I also think the paper needs to explain how the results of their strategy can be used and how they should not be used (I think there is a room for the misinterpretation of such results). A targeted selection of cases, which the clustering analysis implies, means that the results are no random selection of data that can be used in traditional projections. However, such results are useful for case studies, scenarios and in stress testing, and the strategy enables the establishment of a catalogue of weather events with more events than traditional simulations. These points could be made in the paper (in the Discussion).

I also found the paper a bit hard to read and digest, and the figure and table captions especially cryptic. The paper seems to be written for scholars who already are well-versed in the matter, but is less accessible for the wider community. Hence, the paper could benefit from rephrasing some sentences. I hope I have not misunderstood too much of the text.

Some of the caveats are connected with statistics and need at least some discussion. The observations consisted in gridded daily precipitation (REGNIE), but such products are associated with spatial inhomogeneity: because of small-scale features in precipitation, the amount recorded in neighbouring rain gauges are rarely as extreme as each other, which means that the gridded values which are a weighted sum of a number of rain gauge records tend to reduce the extreme values. Moreover, the individual gridded values tend to have a different statistical distribution to the individual underlying rain gauge data (which can be approximated as a gamma distribution). Furthermore, models with different resolution (grid box area) are expected to produce data with different statistical characteristics (area mean) which are not directly comparable to observations (the closest is reanalyses). A related caveat is that a comparison between the area mean from different data sets with different resolutions implies comparing statistical samples of different size, which also are expected to differ merely because of the different sample sizes. To make this even more complicated, the models may generate grid boxes with greater inter-dependency than the observations and less real degrees of freedom. I think such caveats must at least be discussed in the paper, even if it is harder to find a good solution to avoid such shortcomings.

I found a number of statements both in the introduction and on page 19 with which I strongly disagree and think are misconceptions. One reason may be the narrow and biased review of the literature. First of all, statistical downscaling is a term that spans a wide range of techniques, and there have been some examples of poor exercise of statistical downscaling that have given it a bad name. Furthermore, the paper uses a false dichotomy between statistics and physics, which I find unfortunate - but this is also a common misconception.

While there are some types of statistical downscaling techniques which are just statistics (e.g. the analog model, neural nets), there are also statistical downscaling methods which are based on physical dependencies (e.g. regression-based techniques). I have emphasised the importance to use physics as a basis for statistical downscaling in a text book on statistical downscaling [1]. The passage 'the lack of a physical basis behind standard statistical downscaling techniques' is therefore a gross generalisation that is both misleading and incorrect.

While the sentence 'Widely used univariate approaches do not capture physical and spatial dependencies and thus physical and spatial coherence between different meteorological variables may not be maintained after downscaling (Maraun et al., 2010), leading to combinations which are suboptimal as boundary conditions for hydrological modelling' gives a false impression about the merit of statistical downscaling. It is important to stress that the statistical downscaling approach is tailored to a specific use to a much greater degree than dynamical downscaling, and if there has not been a need

to preserve the physical and spatial dependencies, then univariate approaches are adequate. I think this part of the discussion suffers from a limited and biased literature review, as it is perfectly possible to use statistical downscaling for cases where spatial coherence between different meteorological variables is preserved [2]. Furthermore, the regional climate models also suffer from similar problems: (a) when they produce different precipitation patterns to the driving global models, the two levels of models are mutually physically inconsistent, and (b) when the the global and regional circulation models use different parameterisation schemes, they are physically inconsistent. In addition, the regional models tend to produce a smoother picture of the geographical patterns, partly due to the way the lower boundary is provided.

The notion of stationarity (p.2, L.15) is a problem for all models, and the passage 'in the absence of a physical foundation there is no intrinsic reason why a statistical downscaling method which performs well in the present climate should also perform well in a future climate' is a bit like shooting oneself in the foot (keeping in mind that the proposed strategy also makes use of large-scale predictors on par with statistical downsclaing) - in addition to being incorrect (statistical downscaling does not lack a physical foundation in general). All the general circulation models make use of parameterisation schemes (ironically called 'model physics') which essentially are ways to calculate bulk effect of various (unresolved) processes with the help of statistical models (the parameterisation schemes are upscaling rather than downscaling models). Whereas the degree of non-stationarity between scales can be examined in statistically downscaled results, it's much harder in dynamical downscaling and the global models where errors feed back into to model framework with a non-linear effect.

I also find the notion 'statistical downscaling method which performs impressively in one region or season may not work as well in other seasons or regions' somewhat misleading. There is no reason why one would use the same statistical downscaling approach everywhere, but it should instead be tailored to the specific problem. Furthermore, statistical downscaling models should be properly evaluated wherever and

whenever they are applied (there have been poor studies where this has not been done properly). I can use my statistical downscaling framework over the whole world without problem, depending on the availability of good ground observations, but the models need to be tailored to the specific region. Moreover, statistical downscaling has an advantage over dynamical downscaling through low computational costs which makes it ideal for downscaling large multi-model ensembles of global climate model simulations [4]. The small ensemble size of independent dynamically downscaled results is major problem that is likely to produce misleading results according to the law of small numbers, even if the downscaling models themselves were perfect. It is therefore important to stress the need for both statistical and dynamical downscaling. The introduction of the paper and page 19 need a major revision with updated information. It is important to stop the spread of common misconceptions about both statistical and dynamical downscaling.

Minor details:

The concept of added-value is tricky and context-dependent (p.2, L. 20). At least, it needs to be defined, however, more details is not the same as added value. There have been criticism of regional climate models for the lack of added-value [3].

It's a bit of a stretch to use the term "extreme" (and 'PED') for the 99-percentile of rainfall applied to all days: that translates to 3-4 events per year. The label 'heavy rainfall' is more appropriate. (p. 5, L. 1)

Caption of Fig 1 is not easy to understand. Can it be improved?

I found line 30 on page 6 (p.6, L30) a bit cryptic and suggest rephrasing.

Please state the 'pan-European EURO-Cordex domain' (p.7, L-8). It will save the reader looking it up and it should not take much space in the text.

I think that 'internal solutions' is a more appropriate term than 'error growth' (p. 11, L.8) if I have understood the text correctly (the regional model can generate its own description of internal details which may differ from the GCM simulations used for boundary conditions?).

Table 2. Caption is not very helpful, and exactly what does 'All Days' mean?

What is 'this' referring to on p. 12 L.8 ('... is far removed from this as . . .').

Reference to Fig 5 & Fig 1 (p.12, L.13). The ECDF presented is for an area mean precipitation? Please state how many grid boxes/rain gauge stations this statistics comprises. The reason is that aggregated statistics such as sums and averages converge towards a normal distribution ('$\sim N()$') with larger samples. If the obs and CCLM area estimates involve different degrees of freedom (sample size), then we should expect to see different types of curves. It would be easier to interpret these results if information of the number of grid-boxes were provided with some test results on the type of distribution (e.g. Kolmogorov-Smirnov against gamma & $N()$).

I suggest splitting the Summary and Conclusions into a Discussions section and a short conclusions section. This is useful for scholars who browse papers to see if it is of relevance and to make the take-home message clearer.

**Supplement:**

[revised manuscript text omitted]

---

## Author Comment (AC1) · 10 Mar 2018

**Response to Reviewer comments on "A classification algorithm for selective dynamical downscaling of precipitation extremes".**

EP Meredith, HW Rust, U Ulbrich

March 10, 2018

**1   Preliminaries**

We would like to start off by thanking both reviewers for the time and effort they put into reviewing our manuscript. This is most appreciated. Both reviews have raised a number of important points which we agree will improve the manuscript.

In the following pages we set out in detail our responses to the comments of each reviewer and how we plan to act on them.

**2 Response to Reviewer #1 (P Laux)**

*The manuscript presents a very interesting contribution to combine dynamical downscaling approaches with a statistical classification procedure in order to save computational costs. The approach aims at extreme precipitation events and is restricting the dynamical downscaling to those days, in which the probability of extreme events is enhanced. For this reason, the concept of Potential Extreme Days (PEDs) is introduced, which is based on a classification approach of synoptic circulation patterns. The manuscript is well written and understandable in general. The procedure is scientifically sound and clearly described. However, there are concerns in terms of its "applicability" and "usefulness". In order to deserve publication, the following aspects need to be considered and elaborated.*

*My main points center around the efforts required to restrict the dynamical downscaling (in convection-permitting resolution) to selected events only and the credibility of the results obtained:*

*- First, I do not see clearly a potential application behind (at least it is not clearly described in the manuscript). Please elaborate clearly which kind of research and practical application can be considered with this in hydrological modelling.*

*- In addition, it might be difficult for hydrological models to deal with non-continuous data (time series) focusing on the extreme events only. In particular, issues may arise in calibration/validation of such process-based hydrological models based on extreme precipitation events only, i.e. the credibility might be limited if these models are calibrated based on extremes exclusively.*

We recognise the merit of these two points. Indeed, they have similarities with points made by the other reviewer about how we should more clearly explain how results from a procedure like ours should/should not be used, so this was clearly an oversight on our part. We shall include such a discussion in the updated version. As it happens, results generated with this procedure are currently being applied in a number of research sites across Europe, within the framework of the H2020 project BINGO (`http://www.projectbingo.eu/`).

*- The efforts of the classification to identify the PEDs are high. The results depend on the selected domain, number of clusters, selected predictors, selected threshold values, etc. It seems that this is not as straight-forward and to be implemented as described in the manuscript. For instance, a predictor screening must be undertaken if the approach is transferred to other regions. Please elaborate and discuss further.*

We agree that a screening of factors such as predictor variables, thresholds, etc., must be performed before applying the approach to different catchments. It was not our intention to suggest that the method can be directly transferred without modification to other catchments (we even apply it differently in summer and winter). Rather, what we claim can be applied to other catchments is the same methodological framework, subject to necessary changes in predictor variables, thresholds, etc. We can of course not issue blanket guidelines as to how users make these choices, but rather issue guidance as to what would represent best practice. This has already been included for the clustering domain size (P5, L15-17) and number of clusters (P6, L2-7), with a clear objective procedure for the latter. For the choice of variables and thresholds, we already say that these must be empirically obtained (P7, L5-6) and that the choice is flexible (P6, L18-20). Further reflections on the choices of variables/thresholds were also included in the summary (P18, L29-34 / P19, L1-4).

In the revised manuscript, we shall use Sect. 2.2 to further elaborate on the key principles to keep in mind when choosing the predictor variables and thresholds. We shall also endeavor to make it clearer throughout the manuscript that (i) our predictor/threshold "settings" should not be blindly transferred to another catchment, and (ii) that the predictor/threshold "settings" we use may at most be considered as a starting point for applying the algorithm to other catchments. Additionally, since we first submitted our manuscript a new study has been published which provides some guidance for the selection of variables from coarser models which may well-predict

intense precipitation in high-resolution models (Chan et al., 2018); we will cite this too.

> *- From regional climate modelling perspectives, I have concerns in selecting single days only instead of performing continuous simulations. I am referring to the initial conditions, when a new simulation is initiated. It is well-known that these are rather imperfect. This is less problematic for the atmospheric compartment of the RCMs (because of the relatively short memory), however, the terrestrial compartments such as e.g. soil moisture need a certain time to reach equilibrium. For this reason, spin-up periods of several days to weeks might be necessary, which limits the benefit of the presented approach tremendously. In addition to that, time requirements to set-up and submit and control multiple short-term simulations are high.*

We agree with the spin-up point too. For clarity: the 12 km resolution simulations from which the high-resolution (2 km) downscaling is performed are continuous multi-decadal runs, so at the 12 km scale the soil moisture and temperature can safely be considered to be fully spun-up. At the spatial scale of the high-resolution model, however, soil moisture/temperature may not necessarily be in equilibrium. While many studies show that the transient boundary conditions are of first order importance for the occurrence of extreme precipitation (e.g. Pan et al., 1999)), the role of surface feedbacks cannot be discounted in all applications. In the revised version we shall therefore explicitly mention this as a potential limitation for certain applications.

Regarding the time requirements to set up, submit and control multiple short-term simulations, it was not our experience that this was burdensome; in fact it was more-or-less automated via a collection of scripts. Either way, it is incumbent on modellers to carry-out their own cost-benefit analysis prior to implementing any modelling strategy, based on their own needs and resources. We feel that readers are best placed to decide for themselves whether this particular modelling strategy is compatible with their available time/resources.

> *- The application of the classification for the past is well justified, however, it might be very limited for the future ("stationarity" assumption). As correctly mentioned, it can be expected that certain extremal circulation patterns change or other patterns might become more important for extreme events. This is more likely for periods in the far future, e.g. the time slice towards the end of this century, as used in this study. For periods in the far future, I would trust more to the pure dynamical downscaling.*

We fully agree with this comment. Our method cannot *guarantee* that 100 % of the precipitation extremes will be captured, in particular in a future changed climate where new circulation patterns may also cause extreme precipitation. We had thus attempted to highlight this issue in our manuscript (P16, L4-6 and L15-18) and make clear what type of conclusions can and cannot reasonably be drawn after applying the method (P16, L1-11). A similar comment to the above was also made by the other reviewer, who said we need to make clear how the results can/cannot be used in order to avoid misinterpretation. In light of the comments of both reviewers, it is obvious that we did not discuss this issue thoroughly enough. In the revised manuscript we shall aim to better address these issues in an extended discussion.

> *Another concern is the validation of the identified PEDs (section 3.2). I would suggest to check not only the occurrence frequencies, but also the timing of the extremes using the reanalyses data. These can be checked with the timing of the extremes based on station data for the catchment. The frequency is not a good performance criterion to my opinion.*

We're not sure that we fully understand what is being suggested here. As we read it, and especially with the use of the word "timing", it looks like it is being suggested that we compare the PEDs identified from reanalysis with those identified from the 12 km CCLM simulations (ERA-Interim driven) in order to see if the PEDs are occurring on the same days, e.g. if 12.06.2002 is a PED in the reanalysis, is it also a PED in the 12 km CCLM simulations (ERA-Interim driven)? If this is indeed the suggestion, we unfortunately do not feel that it would enhance the analysis.

Although an RCM may be forced at the lateral boundaries by reanalysis, the dynamical downscaling does not

produce a higher-resolution version of that same reanalysis. RCMs without interior contstraints (i.e. some form of internal nudging) are not capable of synchronously reproducing the local-scale (i.e. observed or grid-box) day-to-day variability of their driving model (reanalysis or GCM) (e.g. Maraun and Widmann, 2015); due to the chaotic nature of atmospheric dynamics, small-scale deviations from the driving model will propogate and grow, leading to potentially quite different weather-system trajectories (ibid.). This is particularly true for RCMs with larger, continental-scale, domains: such RCMs exhibit much stronger internal variability (Lucas-Picher et al., 2008). Further still, RCM internal variability – that is, the development of internal solutions divergent from the parent model – is greatest with distance from the inflow boundary and in situations with a weak zonal forcing (Giorgi and Bi, 2000), e.g. summer or blocking situations. Weak zonal forcing, in the Wupper catchment at least, is the large-scale feature most associated with intense convective events.

While the aforementioned does not negate the usefulness of RCM downscaling – taken over sufficiently long time periods the statistics at a given location should still be representative (Maraun and Widmann, 2015) – it does instruct us that one should not a priori expect that PEDs obtained from reanalysis will also be found in RCM simulations forced at the lateral boundaries by that same reanalysis. Indeed, this is very much the case for our study catchment, which is centrally located in the EURO-CORDEX domain. As described in Sect. 3.2 and shown in Table 2, we investigate the coincidence rates between observed extremes and reanalysis-based PEDs, before applying the method to ERA-Interim-forced CCLM simulations and then downscaling the CCLM-based PEDs to 2 km resolution. As the 12 km CCLM simulations are driven with ERA-Interim reanalysis, one might expect that there is a very close temporal coincidence between the PEDs from ERA-Interim and those identified from the 12 km CCLM simulations. As mentioned on page 11 (L2-9), this is however not the case: Only 123 of the 320 PED dates in summer and 150 of 220 PED dates in winter are coincident between ERA-Interim and CCLM. While we briefly sought to explain the reasons for this difference (P11, L3-11), we shall attempt to give further clarity in a revised version. This also relates to comments of the other reviewer, who felt that some sentences could benefit from rephrasing in order to make the paper more accessible to the wider community.

We will thus re-word the relavant text (currently P11, L3-11) along the following lines (subject to minor edits):

This is attributable to the fact that RCMs without interior constraints (i.e. some form of internal nudging) cannot synchronously reproduce the local-scale day-to-day variability of their parent model (Maraun and Widmann, 2015). RCMs of sufficiently large domain size thus often generate large internal variability (e.g Lucas-Picher et al., 2008), often comparable to that found in GCMs (Christensen et al., 2001), which can cause the local RCM solution to significantly deviate from that of its parent model. In the presence of a stronger zonal throughflow, e.g. in winter, the growth of differing internal solutions is limited due to small-scale perturbations being more rapidly swept out of the domain (Giorgi and Bi, 2000). This explains the higher fraction of common days which we find in winter (150/220).

*Minor issues:*

*- I suggest to leave out the code fraction (page 8)*

We are a bit surprised by this comment, as the feedback we received prior to submission was that this code-schematic concisely summarized the method and was helpful for understanding the algorithm. As this has only been categorized as a "minor issue" and the other reviewer has not expressed a similar view, if the Editor agrees we would strongly prefer to retain the schematic in the manuscript.

*- Section 3.3 (Page 13): The authors claim that they perform a performance testing on continuous simulations, but the tests are restricted to the summer periods. I also understood that the RCM downscaling is done only for the summer periods, but maybe I misunderstood this. Anyway, I think it is confusing and the term "continuos" should be omitted.*

Thank you for raising this. The 12 km simulations from Sect. 3.3 are truly continuous, i.e. they were run continuously from 1949-2100 by the CLM-Community, as described in the methodology (Sect. 2.4). The high-resolution simulations downscale this run over 30 historical (1970-1999) and 30 future (2070-2099) 5-month time-slices (April-August each year), as stated in Sect. 2.4. We accept that the term "continuous" could be confusing here and will therefore re-formulate the new version with the use of the term "time-slice" as well, where appropriate.

> - *Please check the brackets given after high-resolution data (abstract, line 1, introduction, lines 21 and 23; Page 18, line 2, etc.)*

Will do. Our reason for using the square brackets was so as not to have the same type of bracket twice in the same expression, we thought this would improve readability. In the new version we will simply delete the square brackets.

**3   Response to Reviewer #2 (R Benestad)**

*The paper "A classification algorithm for selective dynamical downscaling of precipitation extremes" by Meredith et. al. presents an interesting strategy for a more efficient and targeted simulations of heavy precipitation with high-resolution convective-permitting regional climate models. They make use of the dependency of local rainfall on the large-scale (synoptic) conditions in terms of circulation patterns, and apply a cluster analysis to distinguish between days when the conditions are right for heavy rainfall and days when heavy rainfall is unlikely. Then they run a high-resolution regional climate model based on the first principles (physics-based) to simulate rainfall for the selected subset. This approach can in a sense be considered as a hybrid between traditional empirical-statistical downscaling and dynamical downscaling, since statistical techniques (clustering) were used to select times for simulations.*

*The analysis presented in Meredith et. al. are in my opinion scientifically sound and this paper merits publication, but there are a number of important caveats and there are a number of statements with which I think are wrong. I also think the paper needs to explain how the results of their strategy can be used and how they should not be used (I think there is a room for the misinterpretation of such results). A targeted selection of cases, which the clustering analysis implies, means that the results are no random selection of data that can be used in traditional projections. However, such results are useful for case studies, scenarios and in stress testing, and the strategy enables the establishment of a catalogue of weather events with more events than traditional simulations. These points could be made in the paper (in the Discussion).*

We agree that there are important caveats to keep in mind when using data generated via a method like ours as, like you say, the data are not equivalent to a random sample so one must be careful not to misuse the data. In fact, we had tried to discuss this in the manuscript (e.g. P16, L4-6 & L15-18) and make clear what sort of conclusions can/cannot be drawn from such data (P16 L1-11). As the other reviewer also made a similar point, it is clear that we have not adequately communicated this aspect of our results. We shall do so in a revised discussion.

*I also found the paper a bit hard to read and digest, and the figure and table captions especially cryptic. The paper seems to be written for scholars who already are well-versed in the matter, but is less accessible for the wider community. Hence, the paper could benefit from rephrasing some sentences. I hope I have not misunderstood too much of the text.*

We will do our best to add more clarity to the figure and table captions, and also rephrase any sentences which appear difficult to digest. To achieve this, we will have the manuscript checked by a colleague who does not specialize in the field, in case we have missed anything.

*Some of the caveats are connected with statistics and need at least some discussion. The observations consisted in gridded daily precipitation (REGNIE), but such products are associated with spatial inhomogeneity: because of small-scale features in precipitation, the amount recorded in neighbouring rain gauges are rarely as extreme as each other, which means that the gridded values which are a weighted sum of a number of rain gauge records tend to reduce the extreme values. Moreover, the individual gridded values tend to have a different statistical distribution to the individual underlying rain gauge data (which can be approximated as a gamma distribution). Furthermore, models with different resolution (grid box area) are expected to produce data with different statistical characteristics (area mean) which are not directly comparable to observations (the closest is reanalyses). A related caveat is that a comparison between the area mean from different data sets with different resolutions implies comparing statistical samples of different size, which also are expected to differ merely because of the different sample sizes. To make this even more complicated, the models may generate grid boxes with greater inter-dependency than the observations and less real degrees of freedom. I think such caveats must at least be discussed in the paper, even if it is harder to find a good solution to avoid such shortcomings.*

We accept that gridded observational datasets have limitations with respect to precipitation extremes, which is unfortunately an unavoidable problem if considering area averages. An extensive discussion of this issue is however beyond the scope of the manuscript. In the revised manuscript, we shall therefore add text to Sect. 2.1 to (a) make readers aware of and (b) briefly explain this issue; references for further reading may be provided. In this context, we shall add a warning that gridded (observed) precipitation datasets may not be suitable for identifying extremes in certain contexts – e.g. if the station density underlying the gridded data is too low in and around the catchment.

Regarding the issue of sample size (e.g. calculating area mean from datasets with different grid-cell resolution), differences in the means resulting from different numbers of grid cells should decrease as the area over which the mean is being taken increases. We will thus urge readers to be cognizant of the "sample size issue" in particular if their catchment is particularly small and there are large differences between the observational and model grid sizes. As requested below under "minor comments", we will also give the numbers of grid cells within the catchment for the cases of REGNIE (753) and CCLM (161). We also expect that such effects will be smaller when comparing not-dissimilar-resolution grid cells (REGNIE = 1 km, CCLM = 2 km).

In addition to that, there's no reason why our method can't be applied with extremal circulation patterns identified from extreme days at a single station, if this better suits the needs of the end users; this will also be mentioned.

> *I found a number of statements both in the introduction and on page 19 with which I strongly disagree and think are misconceptions. One reason may be the narrow and biased review of the literature. First of all, statistical downscaling is a term that spans a wide range of techniques, and there have been some examples of poor exercise of statistical downscaling that have given it a bad name. Furthermore, the paper uses a false dichotomy between statistics and physics, which I find unfortunate - but this is also a common misconception.*

Having reviewed the comments and offending passages, we accept that our phrasing and occasionally imprecise choice of words appear unjustly critical of statistical downscaling, even though this wasn't the intention. We will respond to the specific cases further below.

Regarding the false dichotomy between physics and statistics, this is also a valid criticism, which arises from our loose usage of the term 'physical basis'. It was thus incorrect of us to say that statistical downscaling 'lacks a physical basis' in general, as you highlight. What we instead want to differentiate is the 'first principles' nature of the (unparametrized) physical processes which can be simulated in dynamical models with the 'empirical-physical' relationships used in statistical downscaling (and many parametrization schemes). A relevant difference, for our purposes, is that the former are based on the primitive equations, which should fulfill the stationarity assumption, whereas the latter are based on empirical physical relationships which may not hold in a future climate. A good illustration of this is found in Prein et al. (2017), where a convection-permitting model (CPM) is used to reveal a different relationship between temperature and precipitation intensity in a future climate. We will make this more clear in the revised version and also avoid unjustly singling out statistical downscaling if parametrization schemes are similarly affected.

> *While there are some types of statistical downscaling techniques which are just statistics (e.g. the analog model, neural nets), there are also statistical downscaling methods which are based on physical dependencies (e.g. regression-based techniques). I have emphasised the importance to use physics as a basis for statistical downscaling in a text book on statistical downscaling [1]. The passage 'the lack of a physical basis behind standard statistical downscaling techniques' is therefore a gross generalisation that is both misleading and incorrect.*

As outlined above, we do not wish to discount the physical basis behind many statistical downscaling techniques and will thus modify this passage accordingly.

> *While the sentence 'Widely used univariate approaches do not capture physical and spatial dependencies and*

*thus physical and spatial coherence between different meteorological variables may not be maintained after downscaling (Maraun et al., 2010), leading to combinations which are suboptimal as boundary conditions for hydrological modelling' gives a false impression about the merit of statistical downscaling. It is important to stress that the statistical downscaling approach is tailored to a specific use to a much greater degree than dynamical downscaling, and if there has not been a need to preserve the physical and spatial dependencies, then univariate approaches are adequate. I think this part of the discussion suffers from a limited and biased literature review, as it is perfectly possible to use statistical downscaling for cases where spatial coherence between different meteorological variables is preserved [2]. Furthermore, the regional climate models also suffer from similar problems: (a) when they produce different precipitation patterns to the driving global models, the two levels of models are mutually physically inconsistent, and (b) when the the global and regional circulation models use different parameterisation schemes, they are physically inconsistent. In addition, the regional models tend to produce a smoother picture of the geographical patterns, partly due to the way the lower boundary is provided.*

To avoid giving false impressions we will remove the reference to univariate approaches, in addition to reformulating the passage. In the revised passage we will focus instead on the aims of our method: namely, to generate gridded data at minimal computational expense which could be used, amongst other things, to force a hydrological model. We will eliminate usage of the term "physically coherent" and instead talk in terms of "physical consistence" between the dynamically downscaled variables. While the different downscaled variables from an RCM may not be in *exact* physical balance – e.g. due to numerical discretization, bulk approximations, etc. – we view physical consistence between output variables as an advantage of RCMs, and especially CPMs where convective processes are explicitly simulated. As one moves to complex hydrological models which require multiple input variables as forcing, the physical consistency between these variables becomes an issue (e.g. we consider a day with large amounts of precipitation while there is no cloud cover as not consistent). Although certainly possible, it would become extremely challenging to represent these complex interdependencies between multiple co-existing variables via statistical downscaling. To take a perhaps extreme example, a statistical downscaling technique could easily make sure that there is full cloud cover whenever precipitation occurs, but this should then have knock-on effects on other input variables like radiation, temperature, humidity, pressure, etc.; accounting for such extensive interdependencies would represent a remarkable challenge outside of dynamical downscaling.

*The notion of stationarity (p.2, L.15) is a problem for all models, and the passage 'in the absence of a physical foundation there is no intrinsic reason why a statistical downscaling method which performs well in the present climate should also perform well in a future climate' is a bit like shooting oneself in the foot (keeping in mind that the proposed strategy also makes use of large-scale predictors on par with statistical downsclaing) - in addition to being incorrect (statistical downscaling does not lack a physical foundation in general). All the general circulation models make use of parameterisation schemes (ironically called 'model physics') which essentially are ways to calculate bulk effect of various (unresolved) processes with the help of statistical models (the parameterisation schemes are upscaling rather than downscaling models). Whereas the degree of non-stationarity between scales can be examined in statistically downscaled results, it's much harder in dynamical downscaling and the global models where errors feed back into to model framework with a non-linear effect.*

Stationarity is a limitation for our method and we have highlighted this on P16 (L4-6, L15-18). As mentioned above, we also intend to remove misleading statements about statistical downscaling lacking a physical basis in general. We also agree that stationarity may be an issue for parametrization schemes. We think that the issue of inheriting incorrect climate change signals from GCMs is still an important one, and would thus like to retain mention of it in a revised introduction. In the revised version, we will however avoid unjustly identifying this problem as one unique to statistical downscaling. In the case of CPMs, whose high-resolution allows convective processes to be explicitly simulated and convective parametrizations to be avoided, there are however a number of studies demonstrating their ability to modify the climate-change signal of summertime extreme precipitation inherited from their coarser parent models (e.g. Kendon et al., 2014; Ban et al., 2015). We will retain this information later in the introduction, as we view our use of CPMs as an advantage of the method.

*I also find the notion 'statistical downscaling method which performs impressively in one region or season may not work as well in other seasons or regions' somewhat misleading. There is no reason why one would use the same statistical downscaling approach everywhere, but it should instead be tailored to the specific problem. Furthermore, statistical downscaling models should be properly evaluated wherever and whenever they are applied (there have been poor studies where this has not been done properly). I can use my statistical downscaling framework over the whole world without problem, depending on the availability of good ground observations, but the models need to be tailored to the specific region. Moreover, statistical downscaling has an advantage over dynamical downscaling through low computational costs which makes it ideal for downscaling large multi-model ensembles of global climate model simulations [4]. The small ensemble size of independent dynamically downscaled results is major problem that is likely to produce misleading results according to the law of small numbers, even if the downscaling models themselves were perfect. It is therefore important to stress the need for both statistical and dynamical downscaling. The introduction of the paper and page 19 need a major revision with updated information. It is important to stop the spread of common misconceptions about both statistical and dynamical downscaling.*

We will delete this sentence and, as mentioned above, re-write the relevant passages of the introduction and discussion.

*Minor details:*

*The concept of added-value is tricky and context-dependent (p.2, L. 20). At least, it needs to be defined, however, more details is not the same as added value. There have been criticism of regional climate models for the lack of added-value [3].*

We agree with this comment and see that our current wording is not precise enough. We shall rephrase the offending sentence to something along the lines of: "Importantly, this AV *should not simply be understood as* represent*ing* increased small-scale detail, but *rather* AV at the spatial scale of the driving GCM due to more processes being represented (Torma et al., 2015)."

*It's a bit of a stretch to use the term "extreme" (and 'PED') for the 99-percentile of rainfall applied to all days: that translates to 3-4 events per year. The label 'heavy rainfall' is more appropriate. (p. 5, L. 1)*

Actually the $99^{th}$ percentile is computed as a seasonal statistic, so it would equate to $< 1$ event per year considering each season separately. We shall make this clearer in the text, and also use the term 'heavy rainfall' (or similar) in the text where appropriate.

*Caption of Fig 1 is not easy to understand. Can it be improved?*

This was also a problem prior to submission, and we tried to make it clearer. We shall try again.

*I found line 30 on page 6 (p.6, L30) a bit cryptic and suggest rephrasing.*

We assume you are referring to the sentence: *To account for different model climatologies, the absolute relative humidity values are transformed into multiples of the model's climatological mean prior to assessment.*, which begins on p.6, L31. We will reformulate this, with added explanation.

What we wanted to communicate is that the relative humidity thresholds (absolute values) obtained from the ERA-Interim training data are not transferred identically to the CCLM simulations. This is because of the considerable differences in RH climatologies that exist between models, which are often just artefacts of how the model internally (or in a post-processing stage) computes relative humidity. There are a number of different ways to compute relative humidity, e.g. saturation with respect to ice or water. Additionally, more sophisticated algorithms which switch between computing the saturation with respect to either ice or water depending on the

ambient temperature often differ in the temperature ranges at which the switch (i.e. from saturation wrt. water $\rightarrow$ saturation wrt. ice) is made. For these reasons, it can be difficult to directly compare RH values between different models. We instead apply the RH thresholds obtained from the training data by re-defining them as a function of the model's climatological mean (e.g. if RH threshold is 90 % and clim. mean is 60 %, then we have $\text{RH}_{thresh,ERAI} = 1.5\text{RH}_{clim,ERAI}$). This function is then applied to the climatological mean of the new model (e.g. $\text{RH}_{thresh,CCLM} = 1.5RH_{clim,CCLM}$) in order to get the absolute value of the RH threshold for that model.

*Please state the 'pan-European EURO-Cordex domain' (p.7, L-8). It will save the reader looking it up and it should not take much space in the text.*

Not sure if we fully understand this comment, as the text already states 'pan-European EURO-CORDEX domain' (Jacob et al. always capitalize "CORDEX"). To avoid readers having to look-up the article, we will also include the approximate coordinates (i.e. lat/lon) of the EURO-CORDEX domain in the sentence.

*I think that 'internal solutions' is a more appropriate term than 'error growth' (p. 11, L.8) if I have understood the text correctly (the regional model can generate its own description of internal details which may differ from the GCM simulations used for boundary conditions?).*

While we think that the term "error growth" is reasonable in the case of RCMs with reanalysis as lateral boundary conditions, as is the case in Sect. 3.2, we also find the term "internal solutions" to be an illuminating alternative, so we will modify the text accordingly.

*Table 2. Caption is not very helpful, and exactly what does 'All Days' mean?*

Thank you for pointing this out. We will add more detail to the caption. 'All Days' is supposed to refer to the set which contains all days, i.e. prior to applying our classification method. This should have been in the caption. Its inclusion is intended to help communicate the improvements gained by applying the classification method, as with the red curves in Figure 5. So, taking the top row of Table 2, we can see that in the set of all days that (obviously) 90 % of days have precipitation below the $90^{th}$ percentile, which is our definition for a "redundant day", i.e. one that you definitely don't want to downscale. Meanwhile amongst days classified as PEDs, only 22.5 % of days are "redundant". We'll also properly separate 'PEDs' and 'All Days' by splitting the rightmost column. Similar changes will be made to Table 3.

*What is 'this' referring to on p. 12 L.8 ('... is far removed from this as . . .').*

'this' was intended to refer to the form of the ECDF seen in Figure 5 for the set of all days (red curve). Rewriting the sentence with the actual entity being referred to, in place of the demonstrative 'this', the sentence would have read: 'The form of the ECDF of the observed PEDs, however, is far removed from the form of the ECDF of all days, as the probability is shifted towards more intense precipitation.' We will reformulate the sentence along the following lines:

'The form of the ECDF of the observed PEDs (blue curve), however, is far removed from that of the set of all days (red curve), as the probability is shifted towards more intense precipitation.'

*Reference to Fig 5 & Fig 1 (p.12, L.13). The ECDF presented is for an area mean precipitation? Please state how many grid boxes/rain gauge stations this statistics comprises. The reason is that aggregated statistics such as sums and averages converge towards a normal distribution ('$\sim N()$') with larger samples. If the obs and CCLM area estimates involve different degrees of freedom (sample size), then we should expect to see different types of curves. It would be easier to interpret these results if information of the number of grid-boxes were provided with some test results on the type of distribution (e.g. Kolmogorov-Smirnov against*

*gamma & N()).*

This is an area average over the catchment (as stated in the caption). We will add the number of grid cells to the caption (REGNIE: 1 km resolution/753 cells; CCLM: 2.2 km resolution/161 cells). We think that we can also find the number of stations in and around the catchment, and will do our best to include this too.

Regarding the issue of sample size and degrees of freedom (DOF) ... If all cells are independent then the sample size is equal to the DOF; all cells are however not independent here. The depencence of the cells, and hence the DOF, is determined by the physical process being described: in our case observed and simulated precipitation. Increasing the resolution at which the physical process is recorded increases the number of cells but also increasese the dependence of these cells. We expect the "effective sample size" to be very similar between the different resolutions because the same process (daily precipitation) is being measured. The improved realism of, in particular heavy, rainfall in CPMs (e.g. Kendon et al., 2012) – including the duration and spatial extent – gives us increased confidence that the physical processes are being realistically modelled and that the grid-cell interdependencies are hence also realistic.

More generally, the motivation behind this plot is to show (i) differences between the red curve (all *observed* days) and blue curve (*observed* PEDs), and (ii) *similarities* between the blue curve and green curve (PEDs downscaled from CCLM-0.11°):

(i) The blue and red curves come from the same dataset (REGNIE) and hence have the same number of grid cells. Differences between the blue and red curves cannot therefore be a statistical artefact of differing sample sizes. We rather show that the PED-based subsample (blue) is very different from the full sample ("all days", red).

(ii) The green curve, however, need not necessarily be similar to the blue curve, because they are different days from simulations and observations, respectively. In fact, what we see is that the blue and green curves are similar, which we claim supports our thesis that if the PEDs are skillfully selected that the convection-permitting model will realistically reproduce the observed PED statistics.

The best solution may well be to add more information to the caption of Fig. 5, so that the message we are trying to convey is clearer.

> *I suggest splitting the Summary and Conclusions into a Discussions section and a short conclusions section. This is useful for scholars who browse papers to see if it is of relevance and to make the take-home message clearer.*

We will do this. However, as Section 3 is already called 'Results and Discussion', we will most likely call the new sections (i) 'Further Discussion' and (ii) 'Conclusions'.

> *References:*
>
> [1] *Benestad, Rasmus E., Inger Hanssen-Bauer, and Deliang Chen. 2008. Empirical-Statistical Downscaling. World Scientific. (free copy:* `http: // rcg. gvc. gu. se/ edu/ esd. pdf` *)*
> [2] *Benestad, Rasmus E., Deliang Chen, Abdelkader Mezghani, Lijun Fan, and Kajsa Parding. 2015. "On Using Principal Components to Represent Stations in Empirical-Statistical Downscaling." Tellus A 67 (0).* `https: // doi. org/ 10. 3402/ tellusa. v67. 28326` *.*
> [3] *Benestad, Rasmus. 2016. "Downscaling Climate Information." Oxford Research Encyclopedia of Climate Science; Oxford University Press, Oxford Research Encyclopedia of Climate Science, , July.* `https: // doi. org/ 10. 1093/ acrefore/ 9780190228620. 013. 27` *.*
> [4] *Benestad, Rasmus, Kajsa Parding, Andreas Dobler, and Abdelkader Mezghani. 2017. "A Strategy to Effectively Make Use of Large Volumes of Climate Data for Climate Change Adaptation." Climate Services.*

*https: // doi. org/ 10. 1016/ j. cliser. 2017. 06. 013 .*

Thank you for the references, and in particular the free copy of your book.

**References**

N. Ban, J. Schmidli, and C. Schär. Heavy precipitation in a changing climate: Does short-term summer precipitation increase faster? *Geophys. Res. Lett.*, 42(4):1165–1172, 2015.

S. C. Chan, E. J. Kendon, N. Roberts, S. Blenkinsop, and H. J. Fowler. Large-scale predictors for extreme hourly precipitation events in convection-permitting climate simulations. *Journal of Climate*, 31(6):2115–2131, 2018. doi: 10.1175/JCLI-D-17-0404.1. URL https://doi.org/10.1175/JCLI-D-17-0404.1.

O. Christensen, M. Gaertner, J. Prego, and J. Polcher. Internal variability of regional climate models. *Clim. Dynam.*, 17(11):875–887, 2001.

F. Giorgi and X. Bi. A study of internal variability of a regional climate model. *J. Geophys. Res.-Atmos.*, 105 (D24):29503–29521, 2000.

E. J. Kendon, N. M. Roberts, C. A. Senior, and M. J. Roberts. Realism of rainfall in a very high-resolution regional climate model. *J. Climate*, 25(17):5791–5806, 2012.

E. J. Kendon, N. M. Roberts, H. J. Fowler, M. J. Roberts, S. C. Chan, and C. A. Senior. Heavier summer downpours with climate change revealed by weather forecast resolution model. *Nat. Clim. Change*, 4(7):570–576, 2014.

P. Lucas-Picher, D. Caya, R. de Elía, and R. Laprise. Investigation of regional climate models' internal variability with a ten-member ensemble of 10-year simulations over a large domain. *Clim. Dynam.*, 31(7-8):927–940, 2008.

D. Maraun and M. Widmann. The representation of location by a regional climate model in complex terrain. *Hydrology and Earth System Sciences*, 19(8):3449–3456, 2015. doi: 10.5194/hess-19-3449-2015. URL https://www.hydrol-earth-syst-sci.net/19/3449/2015/.

Z. Pan, E. Takle, W. Gutowski, and R. Turner. Long simulation of regional climate as a sequence of short segments. *Monthly Weather Review*, 127(3):308–321, 1999. doi: 10.1175/1520-0493(1999)127⟨0308:LSORCA⟩2.0.CO;2. URL https://doi.org/10.1175/1520-0493(1999)127<0308:LSORCA>2.0.CO;2.

A. F. Prein, R. M. Rasmussen, K. Ikeda, C. Liu, M. P. Clark, and G. J. Holland. The future intensification of hourly precipitation extremes. *Nature Climate Change*, 7(1):48, 2017.

---

## Author Response (AR1)

**Response to Reviewer comments on "A classification algorithm for selective dynamical downscaling of precipitation extremes".**

EP Meredith, HW Rust, U Ulbrich

May 3, 2018

**1  Preliminaries**

In the following pages we set out in detail the actions we have taken to address the concerns of the reviewers. At the end of the document, a marked-up version of the revised manuscript is appended, highlighting all of the changes to the text. Aside from changes to the text, the form of the rightmost column in Tables 2 and 3 has been modified to make it easier to understand, and Figure 2 has also been updated (Fig. 2 shows the catchment boundaries, important waterways in the region, and the orography of the 0.02° model). The new Figure 2 now additionally shows the locations of precipitation-measuring stations of the German weather service, so that readers have an idea of the underlying station-density behind the gridded observations we use.

In the initial Author Comments (AC1) we provided our initial responses to the reviewer comments (RC1, RC2) and set out our *planned* changes to the manuscript. The full detailed reasoning for changes made, or not made, is therefore not repeated in this document, but rather presented as necessary in an abridged form. See here for AC1: `https://www.hydrol-earth-syst-sci-discuss.net/hess-2017-660/hess-2017-660-AC1-supplement.pdf`.

All references to page/line numbers are for the *new* version of the manuscript.

**2  Response to Reviewer #1 (P Laux)**

*The manuscript presents a very interesting contribution to combine dynamical downscaling approaches with a statistical classification procedure in order to save computational costs. The approach aims at extreme precipitation events and is restricting the dynamical downscaling to those days, in which the probability of extreme events is enhanced. For this reason, the concept of Potential Extreme Days (PEDs) is introduced, which is based on a classification approach of synoptic circulation patterns. The manuscript is well written and understandable in general. The procedure is scientifically sound and clearly described. However, there are concerns in terms of its "applicability" and "usefulness". In order to deserve publication, the following aspects need to be considered and elaborated.*

*My main points center around the efforts required to restrict the dynamical downscaling (in convection-permitting resolution) to selected events only and the credibility of the results obtained:*

*- First, I do not see clearly a potential application behind (at least it is not clearly described in the manuscript). Please elaborate clearly which kind of research and practical application can be considered with this in hydrological modelling.*

*- In addition, it might be difficult for hydrological models to deal with non-continuous data (time series) focusing on the extreme events only. In particular, issues may arise in calibration/validation of such process-based hydrological models based on extreme precipitation events only, i.e. the credibility might be limited if these models are calibrated based on extremes exclusively.*

As mentioned in AC1, these concerns were also shared by the other reviewer. We have thus taken the following steps:

1. Added brief (one sentence) application examples to the introduction (P3 L28)
2. Added a new paragraph to the 'Further Discussion' (P20 L20-32), in which the main applications of our method and how the data could be applied are discussed. This paragraph starts off by outlining appropriate applications for the data produced via our method, before explaining why observations and/or coarse-model data may be sub-optimal for these applications. The paragraph finishes by stating how models could be calibrated and initialized prior to performing simulations. This is in addition to the already-existing paragraph (P20 L33 - P21 L4) which discusses further applications.
3. The Further Discusson section now also contains a warning about how the data should **NOT** be used (P20 L16-19), to avoid the risk that such data are used to draw unjustifiable conclusions.

*- The efforts of the classification to identify the PEDs are high. The results depend on the selected domain, number of clusters, selected predictors, selected threshold values, etc. It seems that this is not as straightforward and to be implemented as described in the manuscript. For instance, a predictor screening must be undertaken if the approach is transferred to other regions. Please elaborate and discuss further.*

As mentioned in AC1, we agree that a screening of factors such as predictor variables, thresholds, etc., must be performed before applying the approach to different catchments; the method should not be directly transferred to other catchments/situations without modification. We have implemented the following changes:

1. Added guidance to Sect. 2.2 about selecting predictors (P7 L9-11), including the citation of a relavant new study from Chan et al. (2018).
2. In the 'Further Discussion' (P20 L1-2), we add that the predictors we use, or those proposed in Chan et al. (2018), may be used as a *starting* point for applying the method elsewhere, but not more.

3. In Table 1 (the table which shows our predictors/thresholds), we add a warning which says that our predictors/thresholds could be used as a starting point for applying the method to other catchments, but that they should not be directly transferred without first considering meteorological characteristics specific to heavy rainfall events at the new catchment.

We feel that these changes help to better communicate our message that the same methodological *framework* can be applied to other catchments, though subject to necessary changes in predictor variables, thresholds, etc. The issue of predictors performing more/less successfully across different regions and/or seasons is unfortunately unavoidable in any empirical-statistical framework (e.g. Volosciuk et al., 2017). This very point is indeed already covered in the 'Further Discussion' (P19 L20 - P20 L2) when we talk about the desirability of users finding predictors "most suitable to their own catchment". This new information about the selection of predictors is in addition to pre-existing guidance (see e.g. P7 L7-9 & L20-21, P19 L19 - P20 L10).

> *- From regional climate modelling perspectives, I have concerns in selecting single days only instead of performing continuous simulations. I am referring to the initial conditions, when a new simulation is initiated. It is well-known that these are rather imperfect. This is less problematic for the atmospheric compartment of the RCMs (because of the relatively short memory), however, the terrestrial compartments such as e.g. soil moisture need a certain time to reach equilibrium. For this reason, spin-up periods of several days to weeks might be necessary, which limits the benefit of the presented approach tremendously. In addition to that, time requirements to set-up and submit and control multiple short-term simulations are high.*

The issue of soil-moisture and soil-temperature spin-up is an important one, and we have clarified the situation as follows (Sect. 2.3, P8 L13-20):

1. Explained that we initialize the convection-permitting model (CPM) by interpolating from the 12 km resolution model (a standard procedure in 'weather-forecast mode')
2. Explained that the soil components of the 12 km model can be considered fully spun-up at the 12 km scale due to the multi-decadal simulations at 12 km resolution
3. Warned that this does not mean that the soil components will be fully spun-up at the scale of the CPM and that CPMs tend to have a drier soil-moisture climatology
4. Warned that our method thus may not be suitable for simulalating precipitation extremes sensitive to local soil-moisture anomalies

In combination with the aforementioned new discussions of what applications our method is appropriate for (P20 L20-32), some of which don't even require initialization of terrestrial components from the CPM, we believe that appropriate caveats are now provided for the reader.

> *- The application of the classification for the past is well justified, however, it might be very limited for the future ("stationarity" assumption). As correctly mentioned, it can be expected that certain extremal circulation patterns change or other patterns might become more important for extreme events. This is more likely for periods in the far future, e.g. the time slice towards the end of this century, as used in this study. For periods in the far future, I would trust more to the pure dynamical downscaling.*

As mentioned in AC1, we fully agree with this comment. To properly emphasize the issue of stationarity to the reader, we have made the following additions to the text:

1. In the Introduction (P2 L16-18), we explain that the predictor-predictand relationships used in both model parametrization schemes and statistical methods may not remain the same in the future.
2. In the Further Discussion (P20 L12-16) we discuss the stationarity assumption in more detail and then re-emphasize (P20 L17-19) that traditional projections can *only* be made with continuous downscaling.

In addition to the pre-existing text in Sect. 3.4 (where we say that only continuous downscaling would work for the case where new circulation patterns cause precipitation extremes in a future climate), we feel that the issue of stationarity has now been comprehensively addressed.

*Another concern is the validation of the identified PEDs (section 3.2). I would suggest to check not only the occurrence frequencies, but also the timing of the extremes using the reanalyses data. These can be checked with the timing of the extremes based on station data for the catchment. The frequency is not a good performance criterion to my opinion.*

In our original response in AC1, we set out in detail why we did not think that this would enhance the analysis, though with the caveat that we may have misunderstood the suggestion. We thus promised to instead include a more detailed discussion of the issue of RCM internal variability in the manuscript and how this affects the relation between potential extreme days (PEDs) identified from reanalysis and PEDs identified from reanalysis-forced RCM simulations. We have thus re-written the relevant paragraph in Sect. 3.2, starting on page 12 line 7, to better explain that the PEDs identified from ERA-Interim reanalysis are not the same as the PEDs we identify from CCLM simulations which use the same reanalysis as lateral boundary conditions. This re-written section also incorporates some re-wordings suggested by the other reviewer. References on RCM internal variability are provided for further reading.

*Minor issues:*

*- I suggest to leave out the code fraction (page 8)*

As explained in AC1, we would prefer to retain this so long as the Editor agrees, as we feel that the code-schematic concisely summarizes the method and is helpful for understanding the procedure.

*- Section 3.3 (Page 13): The authors claim that they perform a performance testing on continuous simulations, but the tests are restricted to the summer periods. I also understood that the RCM downscaling is done only for the summer periods, but maybe I misunderstood this. Anyway, I think it is confusing and the term "continuos" should be omitted.*

Across the manuscript, we have replaced the term 'continuous' with 'seasonal time-slice' where appropriate. For example, P1 L15-16, P10 L1 & L20, P14 L19 & L26, titles of Sect. 2.4 and 3.3. We also emphasize that the $0.02°$-simulations are continuous from April - August and that analysis is restricted to the summer (JJA) months (P10 L14-15).

*- Please check the brackets given after high-resolution data (abstract, line 1, introduction, lines 21 and 23; Page 18, line 2, etc.)*

We have deleted all square brackets.

**3 Response to Reviewer #2 (R Benestad)**

*The paper "A classification algorithm for selective dynamical downscaling of precipitation extremes" by Meredith et. al. presents an interesting strategy for a more efficient and targeted simulations of heavy precipitation with high-resolution convective-permitting regional climate models. They make use of the dependency of local rainfall on the large-scale (synoptic) conditions in terms of circulation patterns, and apply a cluster analysis to distinguish between days when the conditions are right for heavy rainfall and days when heavy rainfall is unlikely. Then they run a high-resolution regional climate model based on the first principles (physics-based) to simulate rainfall for the selected subset. This approach can in a sense be considered as a hybrid between traditional empirical-statistical downscaling and dynamical downscaling, since statistical techniques (clustering) were used to select times for simulations.*

*The analysis presented in Meredith et. al. are in my opinion scientifically sound and this paper merits publication, but there are a number of important caveats and there are a number of statements with which I think are wrong. I also think the paper needs to explain how the results of their strategy can be used and how they should not be used (I think there is a room for the misinterpretation of such results). A targeted selection of cases, which the clustering analysis implies, means that the results are no random selection of data that can be used in traditional projections. However, such results are useful for case studies, scenarios and in stress testing, and the strategy enables the establishment of a catalogue of weather events with more events than traditional simulations. These points could be made in the paper (in the Discussion).*

To make clearer how data produced via our method can and cannot be used, we have added a new paragraph to the 'Further Discussion' section (P20 L11-19). Here we explicitly state that our method cannot be used in the same way as traditional projections, and that *only* continuous downscaling is appropriate for traditional projections. We also discuss issues related to the assumption of stationarity in this paragraph. Additionally, a second paragraph (P20 L20-32) in the 'Further Discussion' discusses the specific type of modelling applications for which our method could be used and why observations and/or coarser model data may be sub-optimal for such applications. Some of the content of these two paragraphs addresses points raised by the other reviewer.

*I also found the paper a bit hard to read and digest, and the figure and table captions especially cryptic. The paper seems to be written for scholars who already are well-versed in the matter, but is less accessible for the wider community. Hence, the paper could benefit from rephrasing some sentences. I hope I have not misunderstood too much of the text.*

We have done our best to add more clarity to the figure and table captions, and to make the text generally more accessible. The captions for figures 1, 2, and 5-9, and tables 1-3 have all been modified to add more detail and explanation. We hope that they are now easier to follow. We have also attempted to make the text more accessible, e.g. by changing certain words and wordings in places. For example, P13 L3-10.

*Some of the caveats are connected with statistics and need at least some discussion. The observations consisted in gridded daily precipitation (REGNIE), but such products are associated with spatial inhomogeneity: because of small-scale features in precipitation, the amount recorded in neighbouring rain gauges are rarely as extreme as each other, which means that the gridded values which are a weighted sum of a number of rain gauge records tend to reduce the extreme values. Moreover, the individual gridded values tend to have a different statistical distribution to the individual underlying rain gauge data (which can be approximated as a gamma distribution). Furthermore, models with different resolution (grid box area) are expected to produce data with different statistical characteristics (area mean) which are not directly comparable to observations (the closest is reanalyses). A related caveat is that a comparison between the area mean from different data sets with different resolutions implies comparing statistical samples of different size, which also are expected to differ merely because of the different sample sizes. To make this even more complicated, the models may generate grid boxes with greater inter-dependency than the observations and less real degrees of freedom. I*

*think such caveats must at least be discussed in the paper, even if it is harder to find a good solution to avoid such shortcomings.*

To address the limitations of gridded datasets for studying precipitation extremes, we have added new text to Sect. 2.1 to make readers aware of these issues (P5 L16 - P6 L6). Here we discuss issues of spatial variability and homogeneity and emphasize the importance of a sufficiently dense observational network underlying the gridded product for the study of extremes. References are provided for further reading, as an extensive discussion is beyond the scope of this paper. We also added extra information specific to the REGNIE dataset which we use (P5 L14-16) and have marked the locations of the precipitation-measuring stations of the German weather service on Figure 2 (which shows the Wupper catchment and surrounding orography).

To address potential issues arising from differing sample sizes, when presenting the methods (see Sect. 2.3, P9 L6-9) and results (see Sect. 3.2, P14 L14-17) we now make users aware that in certain cases statistical properties may differ simply because of different sample sizes (i.e. number of grid cells comprising area mean), in particular for cases of large differences in grid-cell resolution and for small catchments. In the caption for Fig. 5, where ECDFs from CCLM and REGNIE are plotted alongside each other, we now also list the number of grid cells contained within the Wupper catchment for each dataset, so that results can be more easily interpreted. Additionally, when the REGNIE and CCLM data are first presented in the Methods section, their numbers of grid cells contained within the Wupper catchment are also given in the text. See P5 L13 and P8 L14, respectively.

Finally, we have also added a sentence to the Methods (Sect. 2.1, P6 L5-6) saying that extremal circulation patterns could also be identified from extreme precipitation days taken from a single station, if this station is known to be broadly representative. This would be imaginable for smaller-sized catchments.

> *I found a number of statements both in the introduction and on page 19 with which I strongly disagree and think are misconceptions. One reason may be the narrow and biased review of the literature. First of all, statistical downscaling is a term that spans a wide range of techniques, and there have been some examples of poor exercise of statistical downscaling that have given it a bad name. Furthermore, the paper uses a false dichotomy between statistics and physics, which I find unfortunate - but this is also a common misconception.*

We have removed all references to statistical downscaling which characterize it as lacking a physical basis. We have also added discussions of the strengths/weaknesses of model parametrization schemes, to add more balance to the literature review (specific examples will be referenced further below). Additionally, an effort has been made to somewhat merge the discussions of strengths and weaknesses of statistical and dynamical downscaling methods, so that any comparable weaknesses are presented together, rather than appearing to single-out one particular method (e.g. P2 L5-23).

> *While there are some types of statistical downscaling techniques which are just statistics (e.g. the analog model, neural nets), there are also statistical downscaling methods which are based on physical dependencies (e.g. regression-based techniques). I have emphasised the importance to use physics as a basis for statistical downscaling in a text book on statistical downscaling [1]. The passage 'the lack of a physical basis behind standard statistical downscaling techniques' is therefore a gross generalisation that is both misleading and incorrect.*

As mentioned, we do not wish to discount the physical basis behind many statistical downscaling techniques and have thus deleted all references to statistical downscaling lacking a physical basis.

> *While the sentence 'Widely used univariate approaches do not capture physical and spatial dependencies and thus physical and spatial coherence between different meteorological variables may not be maintained after downscaling (Maraun et al., 2010), leading to combinations which are suboptimal as boundary conditions for hydrological modelling' gives a false impression about the merit of statistical downscaling. It is important to stress that the statistical downscaling approach is tailored to a specific use to a much greater degree than*

*dynamical downscaling, and if there has not been a need to preserve the physical and spatial dependencies, then univariate approaches are adequate. I think this part of the discussion suffers from a limited and biased literature review, as it is perfectly possible to use statistical downscaling for cases where spatial coherence between different meteorological variables is preserved [2]. Furthermore, the regional climate models also suffer from similar problems: (a) when they produce different precipitation patterns to the driving global models, the two levels of models are mutually physically inconsistent, and (b) when the the global and regional circulation models use different parameterisation schemes, they are physically inconsistent. In addition, the regional models tend to produce a smoother picture of the geographical patterns, partly due to the way the lower boundary is provided.*

We have deleted the passage on univariate approaches and physical/spatial coherence. All references to 'coherence' have been removed. As part of a general discussion of RCM added value (P2 L24 - P3 L2), we now mention that RCMs can provide a large set of physically-consistent variables (i.e. consistent amongst the downscaled variables) as input for hydrological models and illustrate with an example what we mean by this (P2 L35 - P3 L2), similar to the example given in Author Comments 1 (AC1).

*The notion of stationarity (p.2, L.15) is a problem for all models, and the passage 'in the absence of a physical foundation there is no intrinsic reason why a statistical downscaling method which performs well in the present climate should also perform well in a future climate' is a bit like shooting oneself in the foot (keeping in mind that the proposed strategy also makes use of large-scale predictors on par with statistical downsclaing) - in addition to being incorrect (statistical downscaling does not lack a physical foundation in general). All the general circulation models make use of parameterisation schemes (ironically called 'model physics') which essentially are ways to calculate bulk effect of various (unresolved) processes with the help of statistical models (the parameterisation schemes are upscaling rather than downscaling models). Whereas the degree of non-stationarity between scales can be examined in statistically downscaled results, it's much harder in dynamical downscaling and the global models where errors feed back into to model framework with a non-linear effect.*

As mentioned in AC1, stationarity is a limitation for our method, which we had previously attempted to highlight (Sect 3.4, P18 L1-3 L12-, P19 L1-2). In the Introduction, we now mention that stationarity of the predictor-predictand relationship in model parametrization schemes and statistical methods cannot be guaranteed (P2 L16-18). We have also added a passage to the 'Further Discussion' (P20 L12-16) explaining how our method could be affected in the absence of stationarity. Here we also mention that stationarity issues are common to both model paramterizations and statistical methods.

*I also find the notion 'statistical downscaling method which performs impressively in one region or season may not work as well in other seasons or regions' somewhat misleading. There is no reason why one would use the same statistical downscaling approach everywhere, but it should instead be tailored to the specific problem. Furthermore, statistical downscaling models should be properly evaluated wherever and whenever they are applied (there have been poor studies where this has not been done properly). I can use my statistical downscaling framework over the whole world without problem, depending on the availability of good ground observations, but the models need to be tailored to the specific region. Moreover, statistical downscaling has an advantage over dynamical downscaling through low computational costs which makes it ideal for downscaling large multi-model ensembles of global climate model simulations [4]. The small ensemble size of independent dynamically downscaled results is major problem that is likely to produce misleading results according to the law of small numbers, even if the downscaling models themselves were perfect. It is therefore important to stress the need for both statistical and dynamical downscaling. The introduction of the paper and page 19 need a major revision with updated information. It is important to stop the spread of common misconceptions about both statistical and dynamical downscaling.*

We have deleted the sentence about a statistical downscaling methods performing differently in different regions. We have in general now produced a major re-write of the Introduction and Discussion so that statistical downscaling is no longer misrepresented and/or unfairly singled-out.

*Minor details:*

*The concept of added-value is tricky and context-dependent (p.2, L. 20). At least, it needs to be defined, however, more details is not the same as added value. There have been criticism of regional climate models for the lack of added-value [3].*

We have clarified this sentence so that it is clear that AV must be considered at the spatial scale of the parent model. It now reads as follows: "Importantly, this AV should not simply be understood as representing increased small-scale detail, but rather AV at the spatial scale of the driving GCM due to more processes being represented (Torma et al., 2015)."

*It's a bit of a stretch to use the term "extreme" (and 'PED') for the 99-percentile of rainfall applied to all days: that translates to 3-4 events per year. The label 'heavy rainfall' is more appropriate. (p. 5, L. 1)*

We have clarified that we take $99^{th}$ percentile as a seasonal statistic (P5 L13), so it would equate to $< 1$ event per year considering each season separately. We have also replaced the term 'extreme' with 'heavy rainfall' in many places.

*Caption of Fig 1 is not easy to understand. Can it be improved?*

We have re-worded and hope that it is now clearer.

*I found line 30 on page 6 (p.6, L30) a bit cryptic and suggest rephrasing.*

This has been re-phrased and we hope that it is now clearer. See P7 L22-25.

*Please state the 'pan-European EURO-Cordex domain' (p.7, L-8). It will save the reader looking it up and it should not take much space in the text.*

We have added the approximate coordinates of the EURO-CORDEX domain to the text (P7 L33).

*I think that 'internal solutions' is a more appropriate term than 'error growth' (p. 11, L.8) if I have understood the text correctly (the regional model can generate its own description of internal details which may differ from the GCM simulations used for boundary conditions?).*

We have modified the text accordingly (P13 L8).

*Table 2. Caption is not very helpful, and exactly what does 'All Days' mean?*

We have added more detail and explanation to table captions 2 and 3. We have additionally changed the appearance of the fifth column in tables 2 and 3, so that it is clearer that we are contrasting the fraction of redundant days between the PEDs and the set containing all days.

*What is 'this' referring to on p. 12 L.8 ('... is far removed from this as . . .').*

The sentence has been reworded as promised in AC1 (P13 L18 - P14 L1).

*Reference to Fig 5 & Fig 1 (p.12, L.13). The ECDF presented is for an area mean precipitation? Please state how many grid boxes/rain gauge stations this statistics comprises. The reason is that aggregated statistics such as sums and averages converge towards a normal distribution ('~N()') with larger samples. If the obs and CCLM area estimates involve different degrees of freedom (sample size), then we should expect to see different types of curves. It would be easier to interpret these results if information of the number of grid-boxes were provided with some test results on the type of distribution (e.g. Kolmogorov-Smirnov against gamma & N()).*

The caption states that this is an area average over the catchment. We have added the number of grid cells to the caption too. We have also marked all precipitation-measuring stations of the German weather service on Fig. 2, which is referred to in the caption for Fig. 5 so that readers know where to look if of interest.

More generally, we have rewritten the caption for the plot so that it is clearer what we are aiming to demonstrate, namely (i) differences between the red curve (all *observed* days) and blue curve (*observed* PEDs), and (ii) *similarities* between the blue curve and green curve (PEDs downscaled from CCLM-0.11° to 0.02°). We have also attempted to add a bit more clarity on this to the main text (Sect. 3.2, P13 L18 - P14 L1).

*I suggest splitting the Summary and Conclusions into a Discussions section and a short conclusions section. This is useful for scholars who browse papers to see if it is of relevance and to make the take-home message clearer.*

We have split the 'Summary and Conclusions' into a 'Further Discussion' and a short 'Conclusions' section. As mentioned in AC1, the former section name was chosen in light of the fact that its preceding section is already called 'Results and Disussion'.

[revised manuscript text omitted]